# A Novel Blind Signal Detector Based on the Entropy of the Power Spectrum Subband Energy Ratio

**DOI:** 10.3390/e23040448

**Published:** 2021-04-11

**Authors:** Han Li, Yanzhu Hu, Song Wang

**Affiliations:** 1School of Modern Post, Beijing University of Posts and Telecommunications, Beijing 100876, China; lihan_60@bupt.edu.cn (H.L.); bupt_automation_safety_yzhu@bupt.edu.cn (Y.H.); 2School of Intelligent Equipment, Shandong University of Science and Technology, Taian 271019, China

**Keywords:** power spectrum subband energy ratio, sample entropy, interval entropy, PSER entropy, sample entropy variance, multinomial distribution, signal detection

## Abstract

In this paper, we present a novel blind signal detector based on the entropy of the power spectrum subband energy ratio (PSER), the detection performance of which is significantly better than that of the classical energy detector. This detector is a full power spectrum detection method, and does not require the noise variance or prior information about the signal to be detected. According to the analysis of the statistical characteristics of the power spectrum subband energy ratio, this paper proposes concepts such as interval probability, interval entropy, sample entropy, joint interval entropy, PSER entropy, and sample entropy variance. Based on the multinomial distribution, in this paper the formulas for calculating the PSER entropy and the variance of sample entropy in the case of pure noise are derived. Based on the mixture multinomial distribution, the formulas for calculating the PSER entropy and the variance of sample entropy in the case of the signals mixed with noise are also derived. Under the constant false alarm strategy, the detector based on the entropy of the power spectrum subband energy ratio is derived. The experimental results for the primary signal detection are consistent with the theoretical calculation results, which proves that the detection method is correct.

## 1. Introduction

With the rapid development of wireless communication, spectrum sensing has been deeply studied and successfully applied in cognitive radio (CR). Common spectrum sensing methods [1,2] can be classified as matched filter detection [3], energy-based detection [4], and cyclostationary-based detection [5], among others. Matched-filtering is the optimum detection method when the transmitted signal is known [6]. Energy detection is considered to be the optimal method when there is no prior information of the transmitted signal. Cyclostationary detection is applicable to signals with cyclostationary features [2]. Energy detection can be classified into two categories: time domain energy detection [4] and frequency domain energy detection [7,8]. The power spectrum subband energy ratio (PSER) detector [9] is a local power spectrum energy detection technology. In order to enable PSER to be used in full spectrum detection, a new detector-based PSER is proposed in this paper, whose detection performance is higher than that of time domain energy detection.

In information theory, entropy is a measure of the uncertainty associated with a discrete random variable, and differential entropy is a measure of uncertainty of a continuous random. As the uncertainty of noise is higher than that of the signal, the entropy of the noise is higher than that of the signal, which is the basis of using entropy to detect signals. Information entropy has been successfully applied to signal detection [10,11,12]. The main entropy detection methods can be classified into two categories: the time domain and the frequency domain.

In the time domain, a signal with a low signal-noise ratio (SNR) is annihilated in the noise, and the estimated entropy is actually the entropy of the noise; therefore, the time domain entropy-based detector does not have an adequate detection performance. Nagaraj [12] introduced a time domain entropy-based matched-filtering for primary users (PU) detection and presented a likelihood ratio test for detecting a PU signal. Gu et al. [13] presented a new cross entropy-based spectrum sensing scheme that has two time-adjacent detected data sets of the PU.

In the frequency domain, the spectral bin amplitude of the signal is obviously higher than that of the noise. Therefore, the frequency domain entropy-based detector is widely applied in many fields. Zhang et al. [10,14] presented a spectrum sensing scheme based on the entropy of the spectrum magnitude, which has been used in many practical applications. For example, Jakub et al. [15] used this scheme to assist in blind signal detection, Zhao [16] improved the two-stage entropy detection method based on this scheme, and Waleed et al. [17] applied this scheme to maritime radar network. So [18] used the conditional entropy of the spectrum magnitude to detect unauthorized user signals in cognitive radio networks. Guillermo et al. [11] proposed an improved entropy estimation method based on Bartlett periodic spectrum. Ye et al. [19] proposed a method based on the exponential entropy. Zhu et al. [20] compared the performances of several entropy detectors in primary user detection. In these papers, the mean of the test statistics based on entropy could be calculated by differential entropy, however the variance of the test statistics based on entropy was not given in these papers, that is, its calculation formula was unknown. In detection theory, the variance of test statistics plays a very important role, therefore the detection based on entropy has a major drawback.

In the above references, none of the entropy-based detectors made use of the PSER. The PSER is a common metric used to represent the proportion of signal energy in a single spectral line. It has been extensively applied in the fields of machine design [21], earthquake modeling [22], remote communication [23,24], and geological engineering [25]. The white noise power spectrum bin conforms to the Gaussian distribution, and its PSER conforms to the Beta distribution [9]. Compared with the entropy detector based on spectral line amplitude, the PSER entropy detector has special properties.

This paper is organized as follows. In Section 2, the theoretical formulas of PSER entropy under pure noise and mixed signal case are deduced. The statistical characteristics of PSER entropy are summarized and the computational complexity of the main statistics are analyzed. Section 3 describes the derivation process of the PSER entropy detector under the constant false alarm strategy in detail. In Section 4, experiments are carried out to verify the accuracy of the PSER entropy detector, and the detection performance is compared with other detection methods. Section 5 provides additional details concerning the research process. The conclusions are drawn in Section 6.

## 2. PSER Entropy

PSER entropy takes the form of a classical Shannon entropy. Theoretically, the Shannon entropy can be approximated by the differential entropy; therefore, this method is adopted in the existing entropy-based signal detection methods [10]. However, this method has the disadvantage that the actual value is not the same as the theoretical value in some cases. Therefore, on the basis of analyzing the statistical characteristics of PSER entropy, this paper proposes a new method to calculate PSER entropy without using differential entropy. Firstly, the range of PSER [0,1] is divided into several equally spaced intervals. Then, under pure noise, the PSER entropy and its variance are derived using the multinomial distribution. Under signal mixed with noise, the PSER entropy and its variance are derived using the mixed multinomial distribution.

### 2.1. Probability Distribution for PSER

The signal mixed with additive Gaussian white noise (GWN) can be expressed as
(1)$$s(n)=\left\{ \begin{array}{l} z(n)\;H_{0}\\ x(n)+z(n)\;H_{1} \end{array}\right.\;,\;n=0,1,\cdots ,L-1$$
where L is the number of sampling points; s(n) is the signal to be detected; x(n) is the signal; z(n) is the GWN with a mean of zero and a variance of $\sigma^{2}$; $H_{0}$ represents the hypothesis corresponding to “no signal transmitted”; and $H_{1}$ corresponds to “signal transmitted”. The single-sided spectrum of s(n) is
(2)$$\vec{S}(k)=\sum_{n=0}^{L-1}s(n)e^{-j\frac{2\pi }{L}kn}\;,k=0,1,\cdots ,\frac{L}{2}-1\;.$$where j is the imaginary unit, and the arrow superscript denotes a complex function. The kth line in the power spectrum of s(n) can be expressed as
(3)$$P(k)=\frac{1}{L}|\vec{S}(k){|}^{2}=\frac{1}{L}\left({\left(X_{R}(k)+Z_{R}(k)\right)}^{2}+{\left(X_{I}(k)+Z_{I}(k)\right)}^{2}\right).$$$X_{R}(k)$ and $X_{I}(k)$ represent the real and imaginary parts of the signal, respectively. $Z_{R}(k)$ and $Z_{I}(k)$ represent the real and imaginary parts of the noise, respectively.

The PSER $B_{d,N}(k)$ is defined as the ratio of the sum of the adjacent d bins from the kth bin in the power spectrum to the entire spectrum energy, i.e.,
(4)$$B_{d,N}(k)=\frac{\sum_{l=k}^{k+d-1}P(l)}{\sum_{i=0}^{N-1}P(i)},1\leq d<N-1,k=0,1,\cdots ,N-d\;,$$
where N=L/2, $\sum_{i=0}^{N-1}P(i)$ represents the total energy in the power spectrum and $\sum_{l=k}^{k+d-1}P(l)$ represents the total energy of the adjacent d bins. When there is noise in the power spectrum, it is clear that $B_{d,N}(k)$ is a random variable.

The probability distribution for $B_{d,N}(k)$ is described in detail in [9]. Under $H_{0}$, $B_{d,\;N}(k)$ follows a beta distribution with parameters d,N−d. Under $H_{1}$, $B_{d,N}(k)$ follows a doubly non-central beta distribution with parameters d,N−d and non-centrality parameters ${\lambda^{\prime }}_{k}$, and $\sum_{i=0}^{N-1}\lambda_{i}-\lambda^{\prime }$, i.e., [9] (p. 7),
$$B_{d,N}(k)=\left\{ \begin{array}{l} \beta (d,N-d)\;\;\;\;H_{0}\\ \beta_{d,N-d}({\lambda^{\prime }}_{k},\sum_{i=0}^{N-1}\lambda_{i}-{\lambda^{\prime }}_{k})\;H_{1} \end{array}\right.\;.$$
where $\lambda_{i}=\left(X_{R}^{2}(k)+X_{I}^{2}(k)\right)/\left(N\sigma^{2}\right)$, i.e., the SNR of the kth spectral bin; ${\lambda^{\prime }}_{k}=\sum_{l=k}^{k+d-1}\lambda_{l}$, the SNR of the *d* spectral lines starting from the kth spectral line; $\sum_{i=0}^{N-1}\lambda_{i}-{\lambda^{\prime }}_{k}$ is the SNR of the spectral lines not contained in the selected subband. The probability density function (PDF) for $B_{d,N}(k)$ [9] (p. 7) is
(5)$$f_{B_{d,N}(k)}(x)=\left\{ \begin{array}{l} \frac{x^{d-1}{\left(1-x\right)}^{N-d-1}}{B(d,N-d)}\;\;H_{0}\\ e^{-\left(\delta_{k,1}+\delta_{k,2}\right)}\sum_{j=0}^{\infty }\sum_{l=0}^{\infty }\frac{\delta_{k,1}^{j}\delta_{k,2}^{l}}{j!l!}\frac{{\left(1-x\right)}^{N+l-d-3}x^{{}^{j+d-1}}}{B(j+d,N-d+l)}\;H_{1} \end{array}\right.\;,x\in \left[0,1\right].$$
where $\delta_{k,1}={\lambda^{\prime }}_{k}/2,\delta_{k,2}=\left(\sum_{i=0}^{N-1}\lambda_{i}-{\lambda^{\prime }}_{k}\right)/2$. The cumulative distribution function (CDF) for $B_{d,N}(k)$ [9] (p. 7) is
(6)$$F_{B_{d,N}(k)}(x)=\left\{ \begin{array}{l} I_{x}(d,N-d)\;\;\;H_{0}\\ e^{-\left(\delta_{k,1}+\delta_{k,2}\right)}\sum_{j=0}^{\infty }\sum_{l=0}^{\infty }\frac{\delta_{k,1}^{j}\delta_{k,2}^{l}}{j!l!}I_{x}(j+d,N+l-d)\;\;H_{1} \end{array}\right.\;,x\in \left[0,1\right].$$

The subband used in this paper only contains one spectral bin, i.e., d=1; therefore, $B_{1,N}(k)$ follows the distribution
(7)$$B_{1,N}(k)=\left\{ \begin{array}{l} \beta (1,N-1)\;\;\;H_{0}\\ \beta_{1,N-1}(\lambda_{k},\sum_{i=0}^{N-1}\lambda_{i}-\lambda_{k})\;\;\;\;H_{1} \end{array}\right.\;.$$

The probability density function (PDF) for $B_{1,N}(k)$ is
(8)$$f_{B_{1,N}(k)}(x)=\left\{ \begin{array}{l} (N-1){\left(1-x\right)}^{N-2}\;\;H_{0}\\ e^{-\left(\delta_{k,1}+\delta_{k,2}\right)}\sum_{j=0}^{\infty }\sum_{l=0}^{\infty }\frac{\delta_{k,1}^{j}\delta_{k,2}^{l}}{j!l!}\frac{{\left(1-x\right)}^{N+l-4}x^{{}^{j}}}{B(j+1,N-1+l)}\;H_{1} \end{array}\right.\;,x\in \left[0,1\right].$$
where $\delta_{k,1}=\lambda_{k}/2,\delta_{k,2}=\left(\sum_{i=0}^{N-1}\lambda_{i}-\lambda_{k}\right)/2$.

The cumulative distribution function (CDF) for $B_{1,N}(k)$ is
(9)$$F_{B_{1,N}(k)}(x)=\left\{ \begin{array}{l} I_{x}(1,N-1)\;\;\;H_{0}\\ e^{-\left(\delta_{k,1}+\delta_{k,2}\right)}\sum_{j=0}^{\infty }\sum_{l=0}^{\infty }\frac{\delta_{k,1}^{j}\delta_{k,2}^{l}}{j!l!}I_{x}(j+1,N+l-1)\;\;H_{1} \end{array}\right.\;,x\in \left[0,1\right].$$

For the convenience of description, $B_{1,N}(k)$ is replaced by $X_{k}$.

### 2.2. Basic Definitions

$X_{k}$ is in the range of [0,1]. The range is divided into m intervals at 1/m, i.e., [0,1/m), [1/m,2/m),…, [(m−1)/m,1]. Then the ith interval is [i/m,(i+1)/m), where i=0,1,2⋯,m−1. The probability that $X_{k}$ falls into the ith interval [26] is
(10)$$p_{i}=\int_{i/m}^{\left(i+1\right)/m}f_{X_{k}}\left(x\right)dx\;,$$
where $p_{0}+p_{1}+\cdots +p_{m-1}=1$. $p_{i}$ is called the interval probability.

The PSER values of all spectral lines are regarded as a sequence $X=\left(X_{0},X_{1},\cdots ,X_{N-1}\right)$. Let the number of times the data in X falls into the ith interval be $t_{i}$, which is a random variable, and $t_{i}=0,1,\cdots ,N$, $\sum_{i=0}^{m-1}t_{i}=N$. Let $t=\left(t_{0},t_{1},\cdots ,t_{m-1}\right)$. The frequency of the data in sequence X falling into the ith interval is denoted as $T_{i}$, i.e., $T_{i}=t_{i}/N$, and $\sum_{i=0}^{m-1}T_{i}=1$.

The random variable $Y_{i}$ is equal to $-T_{i}\log T_{i}$, which is called the sample entropy of PSER. The mean of the sample entropy of PSER in the ith interval is $H_{i}=E(Y_{i})\;,$ and $H_{i}$ is the interval entropy. Notice that any two $t_{i}$ are not independent of each other; therefore, any two $Y_{i}$ are not independent of each other as well. Let $Z\left(t;X\right)=\sum_{i=0}^{m-1}Y_{i}$, i.e.,
(11)$$Z\left(t;X\right)=-\sum_{i=0}^{m-1}\frac{t_{i}}{N}\log\left(\frac{t_{i}}{N}\right).$$

Z(t;X) is called sample entropy of PSER. Z(t;X) can be abbreviated as Z(t) or Z.

By the definition of entropy, entropy is a mean, and it has no variance. However, the sample entropy is the entropy of a PSER sequence sample; therefore the sample entropy is a random variable, and it has a mean and a variance. The mean of the sample entropy is
(12)H(m,N)=E(Z),
where m and N are the numbers of intervals and spectral bins, respectively. H(m,N) can be called the total entropy or PSER entropy. The variance of the sample entropy is denoted as Var(Z). In signal detection, E(Z) and Var(Z) are very important, and the calculation of E(Z) and Var(Z) is discussed in the following sections.

### 2.3. Calculating PSER Entropy Using the Differential Entropy

#### 2.3.1. The Differential Entropy for $X_{k}$

Under $H_{0}$, by the definition of differential entropy, the differential entropy for $X_{k}$ is
$$\begin{array}{l} h(B_{1,N}(k))=-\int_{-\infty }^{+\infty }p(x)\log p(x)dx\\ =-\int_{0}^{+\infty }p(x)\log\left[(N-1){\left(1-x\right)}^{N-2}\right]dx\\ =-\int_{0}^{+\infty }p(x)\left[\log(N-1)+(N-2)\log(1-x)\right]dx\\ =-\log(N-1)-(N-1)(N-2)\int_{0}^{1}{\left(1-x\right)}^{N-2}\log(1-x)dx\\ =\frac{N-2}{\left(N-1\right)\ln a}-\log(N-1), \end{array}$$
where a is the base of the logarithm. When a=e,
(13)$$h(B_{1,N}(k))=\frac{N-2}{N-1}-\ln(N-1).$$

#### 2.3.2. PSER Entropy Calculated Using Differential Entropy

According to the calculation process of the spectrum magnitude entropy presented by Zhang in [10] and the equation given in [27] (p. 247), when 1/m→0, the PSER entropy under $H_{0}$ is
(14)$$\begin{array}{l} H(m,N)=-\sum_{i=0}^{m-1}p_{i}\log p_{i}=-\sum_{i=0}^{m-1}f_{X_{k}}\left(x\right)\frac{1}{m}\ln\left(f_{X_{k}}\left(x\right)\frac{1}{m}\right)\\ ≃h(B_{1,N}(k))-\ln\left(\frac{1}{m}\right)=\frac{N-2}{N-1}-\ln\left(\frac{N-1}{m}\right). \end{array}$$If (N−1)/m<e, then H(m,N) is negative.

#### 2.3.3. The Defect of the PSER Entropy Calculated Using Differential Entropy

PSER entropy is the mean of the sample entropy, and the sample entropy is nonnegative, therefore, the PSER entropy is nonnegative too. However, the PSER entropy calculated by Equation (14) is not always nonnegative. Especially when (N−1)/m<e, the PSER entropy is negative. The reason why Equation (14) is not always nonnegative is that the differential entropy is not always nonnegative.

The difference between the real PSER entropy calculated by simulation experiment and that calculated by differential entropy is shown in Figure 1. At least $10^{4}$ Monte Carlo simulation experiments were carried out under N=256 and different values of m.

In Figure 1, the solid line is the PSER entropy calculated by the differential entropy, while the dotted line is the PSER entropy. When m is very large (1/m<0.005), the results are close. However, when m is small (1/m>0.005), the difference between the two methods is large, and even the total entropy calculated by the differential entropy is negative, which is inconsistent with the actual result. Therefore, in this paper a more reasonable method to calculate the PSER entropy is.

### 2.4. PSER Entropy under $H_{0}$

#### 2.4.1. Definitions and Lemmas

Under $H_{0}$, all $X_{k}$ obey the same beta distribution. According to (10), the interval probability in ith interval is
(15)$$p_{i}=\int_{i/m}^{\left(i+1\right)/m}(N-1){\left(1-t\right)}^{N-2}dt={\left(1-\frac{i}{m}\right)}^{N-1}-{\left(1-\frac{i+1}{m}\right)}^{N-1}.$$$p_{i}$ is essentially the area of the ith interval on the probability density map. Figure 2 shows $p_{0}$ and $p_{1}$ when m is 200, and N is 128.

Let ${Τ}_{m,N}=\left\{t=\left(t_{0},t_{1},\cdots ,t_{m-1}\right):t_{i}\in \{0,1,2,\cdots ,N\},t_{0}+\cdots t_{m-1}=N\right\}$. It is a typical multinomial distribution problem that N data fall into m intervals. By the probability formula of multinomial distribution, the probability of $t=\left(t_{0},t_{1},\cdots ,t_{m-1}\right)$ is
(16)$$\mathrm{Pr}(t)=\mathrm{Pr}(t_{0},t_{1},\cdots ,t_{m-1})=\left(\begin{array}{c} N\\ t_{0},t_{1},\cdots ,t_{m-1} \end{array}\right)p_{0}^{t_{0}}p_{1}^{t_{1}}\cdots p_{m-1}^{t_{m-1}},$$
where
$$\left(\begin{array}{c} N\\ t_{0},t_{1},\cdots ,t_{m-1} \end{array}\right)=\frac{N!}{t_{0}!t_{1}!\cdots ,t_{m-1}!},$$
which is the multinomial coefficient. The following lemmas are used in the following analysis.

**Lemma** **1.***If k is a non-negative integer, then* [28] (p. 183)
(17)$$\mathrm{Pr}\left(t_{i}=k,t\in {Τ}_{m,N}\right)=\frac{N!}{k!\left(N-k\right)!}p_{{}_{i}}^{k}{\left(1-p_{i}\right)}^{N-k}.$$

**Lemma** **2.**
*If
j is a non-negative integer, then*
(18)$$\sum_{t\in {Τ}_{m,N}}\mathrm{Pr}\left(t\right)=\sum_{j=0}^{N}\mathrm{Pr}\left(t_{i}=j,t\in {Τ}_{m,N}\right).$$


**Proof.** The left side of Equation (18) is $\sum_{t\in {Τ}_{m,N}}\mathrm{Pr}\left(t\right)=1$. From Lemma 1, the right side of Equation (18) is
$$\sum_{j=0}^{N}\mathrm{Pr}\left(t_{i}=j,t\in {Τ}_{m,N}\right)=\sum_{j=0}^{N}\frac{N!}{j!\left(N-j\right)!}p_{{}_{i}}^{j}{\left(1-p_{i}\right)}^{N-j}=1.$$  ☐

**Lemma** **3.***If
k and
l are a non-negative integer, and
k+l≤N, then* [28] (p. 183)
(19)$$\mathrm{Pr}\left(t_{i}=k,t_{j}=l,t\in {Τ}_{m,N}\right)=\frac{N!}{k!l!\left(N-k-l\right)!}p_{{}_{i}}^{k}p_{{}_{j}}^{l}{\left(1-p_{i}-p_{j}\right)}^{N-k-l}.$$

**Lemma** **4.**
*If
k and
l are a non-negative integer, and
k+l≤N, then*
(20)$$\sum_{t\in {Τ}_{m,N}}\mathrm{Pr}\left(t\right)=\sum_{k=0}^{N}\sum_{l=0}^{N-k}\mathrm{Pr}\left(t_{i}=k,t_{j}=l,t\in {Τ}_{m,N}\right).$$


**Proof.** From Lemma 3, the right side of Equation (20) is
$$\begin{array}{l} \sum_{k=0}^{N}\sum_{l=0}^{N-k}\mathrm{Pr}\left(t_{i}=k,t_{j}=l,t\in {Τ}_{m,N}\right)\\ =\sum_{k=0}^{N}\sum_{l=0}^{N-k}\frac{N!}{k!l!\left(N-k-l\right)!}p_{{}_{i}}^{k}p_{{}_{j}}^{l}{\left(1-p_{i}-p_{j}\right)}^{N-k-l}\\ =\sum_{k=0}^{N}\frac{N!}{k!\left(N-k\right)!}p_{{}_{i}}^{k}{\left(1-p_{i}\right)}^{N-k}\sum_{l=0}^{N-k}\frac{\left(N-k\right)!}{l!\left(N-k-l\right)!}{\left(\frac{p_{j}}{1-p_{i}}\right)}^{l}{\left(\frac{1-p_{i}-p_{j}}{1-p_{i}}\right)}^{N-k-l}\\ =\sum_{k=0}^{N}\frac{N!}{k!\left(N-k\right)!}p_{{}_{i}}^{k}{\left(1-p_{i}\right)}^{N-k}=1. \end{array}$$  ☐

#### 2.4.2. Statistical Characteristics of $T_{i}$

The mean of $T_{i}$ [28] (p. 183) is
(21)$$E(T_{i})=\sum_{j=0}^{N}\mathrm{Pr}\left(t_{i}=j,t\in {Τ}_{m,N}\right)\frac{j}{N}=\sum_{j=0}^{N}\frac{N!}{j!\left(N-j\right)!}p_{i}^{j}{\left(1-p_{i}\right)}^{N-j}\frac{j}{N}=p_{i}.$$

The mean-square value of $T_{i}$ [28] (p. 183) is
(22)$$E(T_{i}^{2})=\sum_{j=0}^{N}\mathrm{Pr}\left(t_{i}=j,t\in {Τ}_{m,N}\right){\left(\frac{j}{N}\right)}^{2}=\frac{Np_{i}^{2}+p_{i}-p_{i}^{2}}{N}.$$

The variance of $T_{i}$ [28] (p. 183) is
(23)$$Var\left(T_{i}\right)=E\left(T_{i}^{2}\right)-E^{2}\left(T_{i}\right)=\frac{p_{i}\left(1-p_{i}\right)}{N}.$$

#### 2.4.3. Statistical Characteristics of $Y_{i}$

For the convenience of description, let
$$h_{N}(l)=-\frac{l}{N}\log\left(\frac{l}{N}\right)\;,l=0,1\cdots N\;.$$

The mean of the $Y_{i}$ is the mean of all the entropy values of $Y_{i}$ in the ith interval, that is,
(24)$$\begin{array}{l} H_{i}=E(Y_{i})=\sum_{j=0}^{N}\mathrm{Pr}\left(t_{i}=j,t\in {Τ}_{m,N}\right)h_{N}(j)\\ =\sum_{j=0}^{N}\left(\begin{array}{c} N\\ j \end{array}\right)p_{i}^{j}{\left(1-p_{i}\right)}^{N-j}h_{N}\left(j\right). \end{array}$$$H_{i}$ is the interval entropy. The following definition is not used to define the interval entropy in this paper:$$H_{i}=E(Y_{i})=-\sum_{j=0}^{N}\mathrm{Pr}\left(t_{i}=j,t\in {Τ}_{m,N}\right)\log\left(\mathrm{Pr}\left(t_{i}=j,t\in {Τ}_{m,N}\right)\right),$$
as this definition is the entropy of all probabilities of $Y_{i}$ in the ith interval.

The mean-square value of $Y_{i}$ is
(25)$$\begin{array}{l} E\left(Y_{i}{}^{2}\right)=\sum_{j=0}^{N}\mathrm{Pr}\left(t_{i}=j,t\in {Τ}_{m,N}\right)h_{N}^{2}(j)\\ =\sum_{j=0}^{N}\left(\begin{array}{c} N\\ j \end{array}\right)p_{i}^{j}{\left(1-p_{i}\right)}^{N-j}h_{N}^{2}(j). \end{array}$$

The variance of $Y_{i}$ is
(26)$$Var\left(Y_{i}\right)=E\left(Y_{i}{}^{2}\right)-E^{2}\left(Y_{i}\right).$$
when i≠j, the joint entropy of two interval is
(27)$$\begin{array}{l} H_{i,j}=E(Y_{i}Y_{j})=\sum_{t\in {Τ}_{n,N}}\mathrm{Pr}\left(t\right)h_{N}\left(t_{i}\right)h_{N}\left(t_{j}\right)\\ =\sum_{k=0}^{N}\sum_{l=0}^{N-k}\mathrm{Pr}\left(t_{i}=k,t_{j}=l,t\in {Τ}_{m,N}\right)h_{N}(k)h_{N}(l)\\ =\sum_{k=0}^{N}\sum_{l=0}^{N-k}\frac{N!}{k!l!\left(N-k-l\right)!}p_{{}_{i}}^{k}p_{{}_{j}}^{l}{\left(1-p_{i}-p_{j}\right)}^{N-k-l}h_{N}(k)h_{N}(l). \end{array}$$
when i=j, $H_{i,i}=E\left(Y_{i}{}^{2}\right)$, i.e., the mean-square value of $Y_{i}$.

#### 2.4.4. Statistical Characteristics of Z(t)

The total entropy with m intervals and N spectral bins is
(28)$$H\left(m,N\right)=E\left(Z\right)=\sum_{t\in {Τ}_{m,N}}\mathrm{Pr}\left(t\right)Z.$$

**Theorem** **1.**
*The PSER entropy is equal to the sum of all the interval entropy, i.e.,*
(29)$$H\left(m,N\right)=\sum_{i=0}^{m-1}H_{i}.$$


**Proof.** According to Lemma 2,
$$\begin{array}{l} H\left(m,N\right)=\sum_{t\in {Τ}_{m,N}}\mathrm{Pr}\left(t\right)Z=\sum_{t\in {Τ}_{m,N}}^{}\left(\mathrm{Pr}\left(t\right)\times \sum_{i=0}^{m-1}h_{N}\left(t_{i}\right)\right)\\ =\sum_{i=0}^{m-1}\sum_{j=0}^{N}\mathrm{Pr}\left(t_{i}=j,t\in {Τ}_{m,N}\right)h_{N}\left(t_{i}\right)\\ =\sum_{i=0}^{m-1}\sum_{j=0}^{N}\frac{N!}{j!\left(N-j\right)!}p_{{}_{i}}^{j}{\left(1-p_{i}\right)}^{N-j}h_{N}\left(t_{i}\right)\\ =\sum_{i=0}^{m-1}H_{i}. \end{array}$$  ☐

**Theorem** **2.**
*The mean of the mean-square value of the sample entropy is equal to the sum of all the joint entropy of two intervals, i.e.,*
(30)$$E\left(Z^{2}\right)=\sum_{i=0}^{m-1}E\left(Y_{i}{}^{2}\right)+2\sum_{i=0}^{m-2}\sum_{j=i+1}^{m-1}H_{i,j}.$$


**Proof.** From Lemmas 2 and 4,
$$\begin{array}{l} E\left(Z^{2}\right)=\sum_{t\in {Τ}_{m,N}}\mathrm{Pr}\left(t\right)Z^{2}=\sum_{t\in {Τ}_{m,N}}^{}\left(\mathrm{Pr}\left(t\right)\times \sum_{i=0}^{m-1}\sum_{j=0}^{m-1}h_{N}\left(t_{i}\right)h_{N}\left(t_{j}\right)\right)\\ =\sum_{t\in {Τ}_{m,N}}^{}\left(\mathrm{Pr}\left(t\right)\times \left(\sum_{i=0}^{m-1}h_{N}^{2}\left(t_{i}\right)+2\sum_{i=0}^{m-2}\sum_{j=i+1}^{m-1}h_{N}\left(t_{i}\right)h_{N}\left(t_{j}\right)\right)\right)\\ =\sum_{i=0}^{m-1}\sum_{t\in {Τ}_{m,N}}^{}\left(\mathrm{Pr}\left(t\right)h_{N}^{2}\left(t_{i}\right)\right)+2\sum_{i=0}^{m-2}\sum_{j=i+1}^{m-1}\sum_{t\in {Τ}_{m,N}}\mathrm{Pr}\left(t\right)h_{N}\left(t_{i}\right)h_{N}\left(t_{j}\right)\\ =\sum_{i=0}^{m-1}E\left(Y_{i}{}^{2}\right)+2\sum_{i=0}^{m-2}\sum_{j=i+1}^{m-1}H_{i,j}. \end{array}$$  ☐

**Theorem** **3.**
*The variance of the sample entropy is*
(31)$$Var\left(Z\right)=E\left(Z^{2}\right)-E^{2}\left(Z\right)=\sum_{i=0}^{m-1}E\left(Y_{i}{}^{2}\right)+2\sum_{i=0}^{m-2}\sum_{j=i+1}^{m-1}H_{i,j}-{\left(\sum_{i=0}^{m-1}H_{i}\right)}^{2}.$$


For the convenience of description, H(m,N) and Var(Z) under $H_{0}$ are replaced by $\mu_{0}$ and $\sigma_{0}^{2}$, respectively.

#### 2.4.5. Computational Complexity

The calculation time of each statistic is mainly consumed in factorial calculation and the traversal of all cases.

Factorial calculation involves two cases: $\frac{N!}{j!\left(N-j\right)!}$ and $\frac{N!}{k!l!\left(N-k-l\right)!}$. They both have a time complexity of O(N).

There are two methods to traverse all cases: the traversal of one selected interval and the traversal of two selected intervals. The corresponding expressions for these two methods are $\sum_{j=0}^{N}\mathrm{Pr}\left(t_{i}=j,t\in {Τ}_{m,N}\right)$ and $\sum_{k=0}^{N}\sum_{l=0}^{N-k}\mathrm{Pr}\left(t_{i}=k,t_{j}=l,t\in {Τ}_{m,N}\right)$, respectively.

The traversal of one selected interval requires that j take all the values from 0 to N. Considering the time spent computing the factorial, its time complexity is $O\left(N^{2}\right)$. Therefore, the time complexity of calculating the interval entropy is $O\left(N^{2}\right)$.

The traversal of two selected intervals requires firstly selection of k spectral bins from all N spectral bins, and then select l spectral bins from the remaining N−k spectral bins. If k and l are fixed, then the time complexity is
$$O\left(N\frac{N!}{k!l!\left(N-k-l\right)!}\right).$$$\sum_{k=0}^{N}\sum_{l=0}^{N-k}\mathrm{Pr}\left(t_{i}=k,t_{j}=l,t\in {Τ}_{m,N}\right)$ requires listing all the combinations of k and l, and its time complexity is
$$N\left(\left(\begin{array}{c} N\\ 0,0,N \end{array}\right)+\left(\begin{array}{c} N\\ 0,1,N-1 \end{array}\right)+\cdots \left(\begin{array}{c} N\\ N,0,0 \end{array}\right)\right)=N3^{N},$$
i.e., $O\left(N3^{N}\right)$. The time complexity of calculating the interval joint entropy $H_{i,j}$ is $O\left(N3^{N}\right)$.

Calculating the total entropy H(m,N) requires computation of all m interval entropies, so its time complexity is $O\left(mN^{2}\right)$.

It takes the most time to calculate the variance of the sample entropy Var(Z). As all m(m−1)/2 interval joint entropies have to be computed, the time complexity of calculating Var(Z) is $O\left(Nm^{2}3^{N}\right)$. In the following experiments, in order to ensure better detection performance, the values of m and N should not be too small, such as m≥500 and N≥256, and therefore the calculation time will be very long.

### 2.5. PSER Entropy under $H_{1}$

#### 2.5.1. Definitions and Lemmas

Under $H_{1}$, $X_{k}$ obey $\beta_{1,N-1}(\lambda_{k},\sum_{i=0}^{N-1}\lambda_{i}-\lambda_{k})$, and different $X_{k}$ have different non-centrality parameters, and therefore the calculation of total entropy and sample entropy variance under $H_{1}$ is much more complicated than that under $H_{0}$. According to Equation (10), the interval probability in ith interval is
(32)$$p_{k,i}=e^{-\left(\delta_{k,1}+\delta_{k,2}\right)}\sum_{j=0}^{\infty }\sum_{l=0}^{\infty }\frac{\delta_{k,1}^{j}\delta_{k,2}^{l}}{j!l!}\left(I_{\left(i+1\right)/m}(j+1,N+l-1)-I_{i/m}(j+1,N+l-1)\right)\;.$$
where $\sum_{i=0}^{m-1}p_{k,i}=1$. The subscript k stands for the label of $X_{k}$. Figure 3 shows $p_{0}$ and $p_{1}$ when m=200, N=128, $\delta_{k,1}=1$, and $\delta_{k,2}=2$.

Under $H_{1}$, different $X_{k}$ obey different probability distributions, therefore this is a multinomial distribution problem under mixture distributions. Let
$${{Τ}^{\prime }}_{m,N}=\left\{t^{\prime }=\left(t_{0,0},\cdots t_{0,m-1},\cdots t_{N-1,0},\cdots t_{N-1,m-1}\right):t_{k,i}\in \{0,1,2,\cdots ,N\},t_{i}=\sum_{k=0}^{N-1}t_{k,i},\sum_{i=0}^{m-1}t_{i}=N\right\},$$
where $t_{k,i}$ is the number of times of $X_{k}$ falls into the ith interval. $t_{i}$ represents the times of all $X_{k}$ falls into the ith interval. In a sample, since $X_{k}$ can only fall into one interval, $t_{k,i}$ only can be 0 or 1, and $\sum_{i=0}^{m-1}t_{k,i}=1$.

By the probability formula of multinomial distribution, the probability when $t^{\prime }=\left(t_{0,0},\cdots t_{0,m-1},\cdots t_{N-1,0},\cdots t_{N-1,m-1}\right)$ is
(33)$$\begin{array}{l} \mathrm{Pr}(t^{\prime })=\left(\begin{array}{c} N\\ t_{0,0},\cdots t_{0,m-1},\cdots t_{k,0},\cdots t_{k,m-1}\cdots t_{N-1,0},\cdots t_{N-1,m-1} \end{array}\right)p_{0,0}^{t_{0,0}}\cdots p_{0,m-1}^{t_{0,m-1}}\cdots p_{k,0}^{t_{k,0}}\cdots p_{k,m-1}^{t_{k,m-1}}\cdots p_{N-1,0}^{t_{N-1,0}}\cdots p_{N-1,m-1}^{t_{N-1,m-1}}\\ =N!p_{0,0}^{t_{0,0}}\cdots p_{0,m-1}^{t_{0,m-1}}\cdots p_{k,0}^{t_{k,0}}\cdots p_{k,m-1}^{t_{k,m-1}}\cdots p_{N-1,0}^{t_{N-1,0}}\cdots p_{N-1,m-1}^{t_{N-1,m-1}}=N!\prod_{k=0}^{N-1}\prod_{i=0}^{m-1}p_{k,i}^{t_{k,i}}. \end{array}$$

The following lemmas are used in the following analysis.

**Lemma** **5.**
*If
j is a non-negative integer, then*
(34)$$\begin{array}{l} \mathrm{Pr}\left(t_{i}=\sum_{k=0}^{N-1}t_{k,i}=j,t^{\prime }\in {{Τ}^{\prime }}_{m,N}\right)\\ =\frac{N!}{\left(N-j\right)!}\sum_{t_{i}=j}^{}p_{0,i}^{t_{0,i}}\cdots p_{k,i}^{t_{k,i}}\cdots p_{N-1,i}^{t_{N-1,i}}{\left(1-p_{0,i}\right)}^{1-t_{0,i}}\cdots {\left(1-p_{k,i}\right)}^{1-t_{k,i}}\cdots {\left(1-p_{N-1,i}\right)}^{1-t_{N-1,i}}. \end{array}$$


**Proof.** $$\begin{array}{l} \mathrm{Pr}\left(t_{i}=\sum_{k=0}^{N-1}t_{k,i}=j,t^{\prime }\in {{Τ}^{\prime }}_{m,N}\right)=\sum_{t^{\prime }\in {{Τ}^{\prime }}_{m,N},j=\sum_{k=0}^{N-1}t_{k,i}}^{}\mathrm{Pr}\left(t^{\prime }\right)\\ =\sum_{t^{\prime }\in {{Τ}^{\prime }}_{m,N},t_{i}=j}^{}\left(\begin{array}{c} N\\ t_{0,0},\cdots t_{0,m-1},\cdots t_{k,0},\cdots t_{k,m-1}\cdots t_{N-1,0},\cdots t_{N-1,m-1} \end{array}\right)p_{0,0}^{t_{0,0}}\cdots p_{0,m-1}^{t_{0,m-1}}\cdots p_{k,0}^{t_{k,0}}\cdots p_{k,m-1}^{t_{k,m-1}}\cdots p_{N-1,0}^{t_{N-1,0}}\cdots p_{N-1,m-1}^{t_{N-1,m-1}}\\ =\sum_{t^{\prime }\in {{Τ}^{\prime }}_{m,N},t_{i}=j}^{}\left(\begin{array}{c} N\\ t_{0,i},\cdots t_{k,i}\cdots t_{N-1,i},N-j \end{array}\right)p_{0,i}^{t_{0,i}}\cdots p_{k,i}^{t_{k,i}}\cdots p_{N-1,i}^{t_{N-1,i}}{\left(1-p_{0,i}\right)}^{1-t_{0,i}}\cdots {\left(1-p_{k,i}\right)}^{1-t_{k,i}}\cdots {\left(1-p_{N-1,i}\right)}^{1-t_{N-1,i}}\times \\ \left(\begin{array}{c} N-j\\ t_{0,0},\cdots t_{0,i-1},t_{0,i+1},\cdots t_{0,m-1},\cdots t_{N-1,0},\cdots t_{N-1,i-1},t_{N-1,i+1}\cdots t_{N-1,m-1} \end{array}\right)\times {\left(\frac{p_{0,0}}{1-p_{0,i}}\right)}^{t_{0,0}}\cdots {\left(\frac{p_{0,i-1}}{1-p_{0,i}}\right)}^{t_{0,i-1}}\\ {\left(\frac{p_{0,i+1}}{1-p_{0,i}}\right)}^{t_{0,i+1}}\cdots {\left(\frac{p_{0,m-1}}{1-p_{0,m-1}}\right)}^{t_{0,m-1}}\cdots {\left(\frac{p_{N-1,0}}{1-p_{N-1,i}}\right)}^{t_{N-1,0}}\cdots {\left(\frac{p_{N-1,i-1}}{1-p_{N-1,i}}\right)}^{t_{N-1,i-1}}{\left(\frac{p_{N-1,i+1}}{1-p_{N-1,i}}\right)}^{t_{N-1,i+1}}\cdots {\left(\frac{p_{N-1,m-1}}{1-p_{N-1,i}}\right)}^{t_{N-1,m-1}}\\ =\frac{N!}{\left(N-j\right)!}\sum_{t_{i}=j}^{}p_{0,i}^{t_{0,i}}\cdots p_{k,i}^{t_{k,i}}\cdots p_{N-1,i}^{t_{N-1,i}}{\left(1-p_{0,i}\right)}^{1-t_{0,i}}\cdots {\left(1-p_{k,i}\right)}^{1-t_{k,i}}\cdots {\left(1-p_{N-1,i}\right)}^{1-t_{N-1,i}}. \end{array}$$  ☐

**Lemma** **6.**
*If
j is a non-negative integer, then*
(35)$$\sum_{t^{\prime }\in {{Τ}^{\prime }}_{m,N}}\mathrm{Pr}\left(t^{\prime }\right)=\sum_{j=0}^{N}\mathrm{Pr}\left(t_{i}=j,t^{\prime }\in {{Τ}^{\prime }}_{m,N}\right).$$


**Proof.** The left side of Equation (35) is $\sum_{t\in {Τ}_{m,N}}\mathrm{Pr}\left(t\right)=1$. From Lemma 5, the right side of Equation (35) is
$$\begin{array}{l} \sum_{j=0}^{N}\mathrm{Pr}\left(t_{i}=j,t^{\prime }\in {{Τ}^{\prime }}_{m,N}\right)\\ =\sum_{j=0}^{N}\frac{N!}{\left(N-j\right)!}\sum_{t_{i}=j}^{}p_{0,i}^{t_{0,i}}\cdots p_{k,i}^{t_{k,i}}\cdots p_{N-1,i}^{t_{N-1,i}}{\left(1-p_{0,i}\right)}^{1-t_{0,i}}\cdots {\left(1-p_{k,i}\right)}^{1-t_{k,i}}\cdots {\left(1-p_{N-1,i}\right)}^{1-t_{N-1,i}}=1. \end{array}$$  ☐

**Lemma** **7.**
*If
g and
l are a non-negative integer, and
g+l≤N, then*
(36)$$\begin{array}{l} \mathrm{Pr}\left(t_{i}=g,t_{j}=l,t^{\prime }\in {{Τ}^{\prime }}_{m,N}\right)=\frac{N!}{\left(N-g-l\right)!}\sum_{t_{i}=g,t_{j}=l}^{}p_{0,i}^{t_{0,i}}\cdots p_{k,i}^{t_{k,i}}\cdots p_{N-1,i}^{t_{N-1,i}}p_{0,j}^{t_{0,j}}\cdots p_{k,j}^{t_{k,j}}\cdots p_{N-1,j}^{t_{N-1,j}}\times \\ {\left(1-p_{0,i}\right)}^{1-t_{0,i}}\cdots {\left(1-p_{k,i}\right)}^{1-t_{k,i}}\cdots {\left(1-p_{N-1,i}\right)}^{1-t_{N-1,i}}{\left(1-p_{0,j}\right)}^{1-t_{0,j}}\cdots {\left(1-p_{k,j}\right)}^{1-t_{k,j}}\cdots {\left(1-p_{N-1,j}\right)}^{1-t_{N-1,j}}. \end{array}$$


**Proof.** $$\begin{array}{l} \mathrm{Pr}\left(t_{i}=g,t_{j}=l,t^{\prime }\in {{Τ}^{\prime }}_{m,N}\right)=\sum_{t^{\prime }\in {{Τ}^{\prime }}_{m,N},t_{i}=g,t_{j}=l}^{}\mathrm{Pr}\left(t^{\prime }\right)\\ =\sum_{t^{\prime }\in {{Τ}^{\prime }}_{m,N},t_{i}=g,t_{j}=l}^{}\left(\begin{array}{c} N\\ t_{0,0},\cdots t_{0,m-1},\cdots t_{k,0},\cdots t_{k,m-1}\cdots t_{N-1,0},\cdots t_{N-1,m-1} \end{array}\right)p_{0,0}^{t_{0,0}}\cdots p_{0,m-1}^{t_{0,m-1}}\cdots p_{k,0}^{t_{k,0}}\cdots p_{k,m-1}^{t_{k,m-1}}\cdots p_{N-1,0}^{t_{N-1,0}}\cdots p_{N-1,m-1}^{t_{N-1,m-1}}\\ =\sum_{t^{\prime }\in {{Τ}^{\prime }}_{m,N},t_{i}=g,t_{j}=l}^{}\left(\begin{array}{c} N\\ t_{0,i},\cdots t_{k,i}\cdots t_{N-1,i},t_{0,j},\cdots t_{k,j}\cdots t_{N-1,j},N-g-l \end{array}\right)p_{0,i}^{t_{0,i}}\cdots p_{k,i}^{t_{k,i}}\cdots p_{N-1,i}^{t_{N-1,i}}p_{0,j}^{t_{0,j}}\cdots p_{k,j}^{t_{k,j}}\cdots p_{N-1,j}^{t_{N-1,j}}\times \\ {\left(1-p_{0,i}\right)}^{1-t_{0,i}}\cdots {\left(1-p_{k,i}\right)}^{1-t_{k,i}}\cdots {\left(1-p_{N-1,i}\right)}^{1-t_{N-1,i}}{\left(1-p_{0,j}\right)}^{1-t_{0,j}}\cdots {\left(1-p_{k,j}\right)}^{1-t_{k,j}}\cdots {\left(1-p_{N-1,j}\right)}^{1-t_{N-1,j}}\\ \left(\begin{array}{c} N-g-l\\ t_{0,0},\cdots t_{0,i-1},t_{0,i+1},\cdots t_{0,j-1},t_{0,j+1},\cdots t_{0,m-1},\cdots t_{N-1,0},\cdots t_{N-1,i-1},t_{N-1,i+1}\cdots t_{N-1,j-1},t_{N-1,j+1}\cdots t_{N-1,m-1} \end{array}\right)\times {\left(\frac{p_{0,0}}{1-p_{0,i}}\right)}^{t_{0,0}}\cdots {\left(\frac{p_{0,i-1}}{1-p_{0,i}}\right)}^{t_{0,i-1}}\\ {\left(\frac{p_{0,i+1}}{1-p_{0,i}}\right)}^{t_{0,i+1}}\cdots {\left(\frac{p_{0,m-1}}{1-p_{0,m-1}}\right)}^{t_{0,m-1}}\cdots {\left(\frac{p_{N-1,0}}{1-p_{N-1,j}}\right)}^{t_{N-1,0}}\cdots {\left(\frac{p_{N-1,i-1}}{1-p_{N-1,j}}\right)}^{t_{N-1,i-1}}{\left(\frac{p_{N-1,i+1}}{1-p_{N-1,j}}\right)}^{t_{N-1,i+1}}\cdots {\left(\frac{p_{N-1,m-1}}{1-p_{N-1,j}}\right)}^{t_{N-1,m-1}}\\ =\sum_{t_{i}=g,t_{j}=l}^{}\left(\begin{array}{c} N\\ t_{0,i},\cdots t_{k,i}\cdots t_{N-1,i},t_{0,j},\cdots t_{k,j}\cdots t_{N-1,j},N-g-l \end{array}\right)p_{0,i}^{t_{0,i}}\cdots p_{k,i}^{t_{k,i}}\cdots p_{N-1,i}^{t_{N-1,i}}p_{0,j}^{t_{0,j}}\cdots p_{k,j}^{t_{k,j}}\cdots p_{N-1,j}^{t_{N-1,j}}\times \\ {\left(1-p_{0,i}\right)}^{1-t_{0,i}}\cdots {\left(1-p_{k,i}\right)}^{1-t_{k,i}}\cdots {\left(1-p_{N-1,i}\right)}^{1-t_{N-1,i}}{\left(1-p_{0,j}\right)}^{1-t_{0,j}}\cdots {\left(1-p_{k,j}\right)}^{1-t_{k,j}}\cdots {\left(1-p_{N-1,j}\right)}^{1-t_{N-1,j}}\\ =\frac{N!}{\left(N-g-l\right)!}\sum_{t_{i}=g,t_{j}=l}^{}p_{0,i}^{t_{0,i}}\cdots p_{k,i}^{t_{k,i}}\cdots p_{N-1,i}^{t_{N-1,i}}p_{0,j}^{t_{0,j}}\cdots p_{k,j}^{t_{k,j}}\cdots p_{N-1,j}^{t_{N-1,j}}\times \\ {\left(1-p_{0,i}\right)}^{1-t_{0,i}}\cdots {\left(1-p_{k,i}\right)}^{1-t_{k,i}}\cdots {\left(1-p_{N-1,i}\right)}^{1-t_{N-1,i}}{\left(1-p_{0,j}\right)}^{1-t_{0,j}}\cdots {\left(1-p_{k,j}\right)}^{1-t_{k,j}}\cdots {\left(1-p_{N-1,j}\right)}^{1-t_{N-1,j}}. \end{array}$$  ☐

#### 2.5.2. Statistical Characteristics of $T_{i}$

The mean of $T_{i}$ is
(37)$$\begin{array}{l} E(T_{i})=\sum_{j=0}^{N}\mathrm{Pr}\left(t_{i}=j,t^{\prime }\in {{Τ}^{\prime }}_{m,N}\right)\frac{j}{N}\\ =\sum_{j=0}^{N}\sum_{t_{i}=j}^{}\left(\begin{array}{c} N\\ t_{0,i},\cdots t_{k,i}\cdots t_{N-1,i},N-j \end{array}\right)p_{0,i}^{t_{0,i}}\cdots p_{k,i}^{t_{k,i}}\cdots p_{N-1,i}^{t_{N-1,i}}{\left(1-p_{0,i}\right)}^{1-t_{0,i}}\cdots {\left(1-p_{k,i}\right)}^{1-t_{k,i}}\cdots {\left(1-p_{N-1,i}\right)}^{1-t_{N-1,i}}\frac{j}{N}\\ =\sum_{j=0}^{N}\frac{j}{N}\frac{N!}{\left(N-j\right)!}\sum_{t_{i}=j}^{}p_{0,i}^{t_{0,i}}\cdots p_{k,i}^{t_{k,i}}\cdots p_{N-1,i}^{t_{N-1,i}}{\left(1-p_{0,i}\right)}^{1-t_{0,i}}\cdots {\left(1-p_{k,i}\right)}^{1-t_{k,i}}\cdots {\left(1-p_{N-1,i}\right)}^{1-t_{N-1,i}}. \end{array}$$

The mean-square value of $T_{i}$ is
(38)$$\begin{array}{l} E(T_{i}^{2})=\sum_{j=0}^{N}\mathrm{Pr}\left(t_{i}=j,t^{\prime }\in {{Τ}^{\prime }}_{m,N}\right){\left(\frac{j}{N}\right)}^{2}\\ =\sum_{j=0}^{N}{\left(\frac{j}{N}\right)}^{2}\frac{N!}{\left(N-j\right)!}\sum_{t_{i}=j}^{}p_{0,i}^{t_{0,i}}\cdots p_{k,i}^{t_{k,i}}\cdots p_{N-1,i}^{t_{N-1,i}}{\left(1-p_{0,i}\right)}^{1-t_{0,i}}\cdots {\left(1-p_{k,i}\right)}^{1-t_{k,i}}\cdots {\left(1-p_{N-1,i}\right)}^{1-t_{N-1,i}}. \end{array}$$

The variance of $T_{i}$ is
$$Var\left(T_{i}\right)=E\left(T_{i}^{2}\right)-E^{2}\left(T_{i}\right).$$

#### 2.5.3. Statistical Characteristics of $Y_{i}$

The mean of the $Y_{i}$ is
(39)$$\begin{array}{l} H_{i}=E(Y_{i})=\sum_{j=0}^{N}\mathrm{Pr}\left(t_{i}=j,t^{\prime }\in {{Τ}^{\prime }}_{m,N}\right)h_{N}(j)\\ =\sum_{j=0}^{N}\frac{j}{N}\frac{N!}{\left(N-j\right)!}h_{N}(j)\sum_{t_{i}=j}^{}p_{0,i}^{t_{0,i}}\cdots p_{k,i}^{t_{k,i}}\cdots p_{N-1,i}^{t_{N-1,i}}{\left(1-p_{0,i}\right)}^{1-t_{0,i}}\cdots {\left(1-p_{k,i}\right)}^{1-t_{k,i}}\cdots {\left(1-p_{N-1,i}\right)}^{1-t_{N-1,i}}. \end{array}$$

The mean-square value of $Y_{i}$ is
(40)$$\begin{array}{l} E\left(Y_{i}{}^{2}\right)=\sum_{j=0}^{N}\mathrm{Pr}\left(t_{i}=j,t^{\prime }\in {{Τ}^{\prime }}_{m,N}\right)h_{N}^{2}(j)\\ =\sum_{j=0}^{N}\frac{j}{N}\frac{N!}{\left(N-j\right)!}h_{N}^{2}(j)\sum_{t_{i}=j}^{}p_{0,i}^{t_{0,i}}\cdots p_{k,i}^{t_{k,i}}\cdots p_{N-1,i}^{t_{N-1,i}}{\left(1-p_{0,i}\right)}^{1-t_{0,i}}\cdots {\left(1-p_{k,i}\right)}^{1-t_{k,i}}\cdots {\left(1-p_{N-1,i}\right)}^{1-t_{N-1,i}}. \end{array}$$

The variance of $Y_{i}$ is
$$Var\left(Y_{i}\right)=E\left(Y_{i}{}^{2}\right)-E^{2}\left(Y_{i}\right).$$
when i≠j, the joint entropy of two interval is
(41)$$\begin{array}{l} H_{i,j}=E(Y_{i}Y_{j})=\sum_{t^{\prime }\in {{Τ}^{\prime }}_{m,N}}\mathrm{Pr}\left(t^{\prime }\right)h_{N}\left(t_{i}\right)h_{N}\left(t_{j}\right)\\ =\sum_{g=0}^{N}\sum_{l=0}^{N-g}\mathrm{Pr}\left(t_{i}=g,t_{j}=l,t^{\prime }\in {{Τ}^{\prime }}_{m,N}\right)h_{N}(g)h_{N}(l) \end{array}$$
when i=j, $H_{i,i}=E\left(Y_{i}{}^{2}\right)$, i.e., the mean-square value of $Y_{i}$.

#### 2.5.4. Statistical Characteristics of $Z\left(t^{\prime }\right)$

Under $H_{1}$, the PSER entropy is
$$H\left(m,N\right)=\sum_{t^{\prime }\in {{Τ}^{\prime }}_{m,N}}\mathrm{Pr}\left(t^{\prime }\right)Z\left(t^{\prime }\right).$$

**Theorem** **4.**
*Under*
$H_{1}$
*, the PSER entropy is equal to the sum of all the interval entropy, i.e.,*
(42)$$H\left(m,N\right)=\sum_{j=0}^{m-1}H_{j}.$$


**Proof.** $$\begin{array}{l} \sum_{t^{\prime }\in {{Τ}^{\prime }}_{m,N}}\mathrm{Pr}\left(t^{\prime }\right)Z=\sum_{t^{\prime }\in {{Τ}^{\prime }}_{m,N}}^{}\left(\mathrm{Pr}\left(t^{\prime }\right)\times \sum_{i=0}^{m-1}h_{N}\left(t_{i}\right)\right)\\ =\sum_{i=0}^{m-1}\sum_{t^{\prime }\in {{Τ}^{\prime }}_{m,N}}^{}\mathrm{Pr}\left(t^{\prime }\right)h_{N}\left(t_{i}\right)=\sum_{i=0}^{m-1}\sum_{j=0}^{N}\mathrm{Pr}\left(t_{i}=j,t^{\prime }\in {{Τ}^{\prime }}_{m,N}\right)h_{N}\left(t_{i}\right)\\ =\sum_{i=0}^{m-1}H_{i}. \end{array}$$  ☐

The mean of the mean-square value of $Z\left(t^{\prime }\right)$ is
(43)$$\begin{array}{l} E\left(Z^{2}\right)=\sum_{t^{\prime }\in {{Τ}^{\prime }}_{m,N}}\mathrm{Pr}\left(t^{\prime }\right)Z^{2}=\sum_{t^{\prime }\in {{Τ}^{\prime }}_{m,N}}^{}\left(\mathrm{Pr}\left(t^{\prime }\right)\times \sum_{i=0}^{m-1}\sum_{j=0}^{m-1}h_{N}\left(t_{i}\right)h_{N}\left(t_{j}\right)\right)\\ =\sum_{i=0}^{m-1}E\left(Y_{i}{}^{2}\right)+2\sum_{i=0}^{m-2}\sum_{j=i+1}^{m-1}H_{i,j}. \end{array}$$

The variance of $Z\left(t^{\prime }\right)$ is
(44)$$Var\left(Z\right)=E\left(Z^{2}\right)-E^{2}\left(Z\right)=\sum_{i=0}^{m-1}E\left(Y_{i}{}^{2}\right)+2\sum_{i=0}^{m-2}\sum_{j=i+1}^{m-1}H_{i,j}-{\left(\sum_{i=0}^{m-1}H_{i}\right)}^{2}.$$

For the convenience of description, H(m,N) and Var(Z) are denoted as $\mu_{1}$ and $\sigma_{1}^{2}$ under $H_{1}$.

#### 2.5.5. Computational Complexity

Under $H_{1}$, many statistics take a much longer time than under $H_{0}$. The calculation time is mainly consumed in two aspects: calculation of $p_{k,i}$, and the factorial calculation and the traversal of all cases.

Calculation of $p_{k,i}$

As seen from Equation (32), $p_{k,i}$ is expressed by infinite double series under $H_{1}$, and its value could only be obtained by numerical calculation. Since the number of calculation terms is set to be large, it will take a significant amount of calculation time.

2.Factorial calculation

Similar to the analysis under $H_{0}$, the time complexity of the factorial calculation is O(N).

3.The traversal of all cases

There are two methods to traverse all cases: the traversal of one selected interval and the traversal of two selected intervals. The corresponding expressions for these two methods are $\sum_{j=0}^{N}\mathrm{Pr}\left(t_{i}=j,t^{\prime }\in {{Τ}^{\prime }}_{m,N}\right)$ and $\sum_{g=0}^{N}\sum_{l=0}^{N-g}\mathrm{Pr}\left(t_{i}=g,t_{j}=l,t^{\prime }\in {{Τ}^{\prime }}_{m,N}\right)$, respectively.

Calculating $\mathrm{Pr}\left(t_{i}=j,t^{\prime }\in {{Τ}^{\prime }}_{m,N}\right)$ is a process of choosing j of N lines, and its time complexity is $O\left(NC_{N}^{j}\right)$. Similar to the analysis under $H_{0}$, the computational complexity of $\sum_{j=0}^{N}\mathrm{Pr}\left(t_{i}=j,t^{\prime }\in {{Τ}^{\prime }}_{m,N}\right)$ is $O\left(N2^{N}\right)$. Therefore, the time complexity of calculating the interval entropy is $O\left(N2^{N}\right)$.

The computational complexity of $\sum_{k=0}^{N}\sum_{l=0}^{N-k}\mathrm{Pr}\left(t_{i}=k,t_{j}=l,t\in {Τ}_{m,N}\right)$ is $O\left(N^{3}3^{N}\right)$. Therefore the time complexity of calculating the interval joint entropy $H_{i,j}$ is $O\left(N^{3}3^{N}\right)$. The time complexity of calculating the PSER entropy is $O\left(mN^{3}3^{N}\right)$. The time complexity of calculating the variance of the sample entropy Var(Z) is $O\left(m^{2}N^{3}3^{N}\right)$.

## 3. Signal Detector Based on the PSER Entropy

In this section, a signal detection method based on PSER entropy is deduced under the constant false alarm (CFAR) strategy according to the PSER entropy and sample entropy variance derived in Section 2. This method is also called full power spectrum subband energy ratio entropy detector (FPSED), because it detects on the full power spectrum.

### 3.1. Principle

Signal detection based on PSER entropy takes the sample entropy as a detection statistic to judge whether there is a signal in the whole spectrum. The sample entropy is
$$Z=\sum_{i=0}^{m-1}Y_{i}$$

The PSER entropy of GWN is obviously different from that of the mixed signal. In general, the PSER entropy of the mixed signal will be less than that of GWN, but sometimes it will also be greater than that of GWN. This can be seen in Figure 4.

In Figure 4a, the PSER entropy of GWN is higher than that of the noisy Ricker signal. However, the PSER entropy of GWN is lower than that of the noisy Ricker signal in Figure 4b. Therefore, when setting the detection threshold of the PSER entropy detector, the relationship between the PSER entropy of the signal and that of noise should be considered.

#### 3.1.1. The PSER Entropy of a Signal Less Than That of GWN

When the PSER entropy of signal is less than that of GWN, let the threshold be $\eta_{1}$ which tests the decision statistic. If the test statistic is less than $\eta_{1}$, the signal is deemed to be present, and it is absent otherwise, i.e.,
(45)$$\left\{ \begin{array}{l} Z>\eta_{1}\;\;H_{0}\\ Z<\eta_{1}\;\;H_{1} \end{array}\right..$$

The distribution of Z is regarded as Gaussian in this paper, so
(46)$$Z∼\{\begin{array}{c} N\left(\mu_{0},\sigma_{0}^{2}\right),H_{0}\\ N\left(\mu_{1},\sigma_{1}^{2}\right),H_{1} \end{array}.$$

Under the CFAR strategy, the false alarm probability $P_{f}$ can be expressed as follows:(47)$$P_{f}=\mathrm{Pr}\left(Z<\eta_{1}\;|H_{0}\right)=1-Q\left(\frac{\eta_{1}-\mu_{0}}{\sigma_{0}}\right).$$$\eta_{1}$ can be derived from Equation (47)
(48)$$\eta_{1}=Q^{-1}(1-P_{f})\sigma_{0}+\mu_{0}.$$

The detection probability $P_{d}$ can be expressed as follows:(49)$$P_{d}=\mathrm{Pr}\left(Z<\eta_{1}\;|H_{1}\right)=1-Q\left(\frac{\eta_{1}-\mu_{1}}{\sigma_{1}}\right).$$Substituting Equation (48) into Equation (49), $P_{d}$ can then be evaluated as follows:(50)$$P_{d}=1-Q\left(\frac{Q^{-1}(1-P_{f})\sigma_{0}+\mu_{0}-\mu_{1}}{\sigma_{1}}\right).$$

#### 3.1.2. The PSER Entropy of a Signal Larger Than That of GWN

When the PSER entropy of a signal is larger than that of GWN, let the threshold be $\eta_{2}$. If the test statistic is larger than $\eta_{2}$, the signal is deemed to be present, and it is absent otherwise, i.e.,
(51)$$\left\{ \begin{array}{l} Z<\eta_{2}\;\;H_{0}\\ Z>\eta_{2}\;\;H_{1} \end{array}\right..$$

The false alarm probability $P_{f}$ is
(52)$$P_{f}=\mathrm{Pr}\left(Z>\eta_{2}\;|H_{0}\right)=Q\left(\frac{\eta_{2}-\mu_{0}}{\sigma_{0}}\right),$$
and
(53)$$\eta_{2}=Q^{-1}(P_{f})\sigma_{0}+\mu_{0}.$$

The detection probability $P_{d}$ is
(54)$$P_{d}=\mathrm{Pr}\left(Z>\eta_{2}\;|H_{1}\right)=Q\left(\frac{\eta_{2}-\mu_{1}}{\sigma_{1}}\right).$$Substituting Equation (53) into Equation (54), $P_{d}$ can then be evaluated as follows:(55)$$P_{d}=Q\left(\frac{Q^{-1}(P_{f})\sigma_{0}+\mu_{0}-\mu_{1}}{\sigma_{1}}\right).$$

### 3.2. Other Detection Methods

In the following experiment, the PSER entropy detector compare with the commonly used full spectrum energy detection (FSED) [29] and matched-filtering detector (MFD) methods, under the same condition. In this section, we introduce these two detectors.

#### 3.2.1. Full Spectrum Energy Detection

The performance of FSED is exactly the same as that of classical energy detection (ED). The total spectral energy is measured by the sum of all spectral lines in the power spectrum, that is,
(56)$$T_{FD}=\sum_{k=0}^{N-1}P(k).$$

Let γ be the SNR, that is, $\gamma =\frac{1}{N^{2}\sigma^{2}}\sum_{k=0}^{N-1}\left(X_{R}^{2}(k)+X_{I}^{2}(k)\right)$. When the detection length N is large enough, $T_{FD}$ obeys a Gaussian distribution:(57)$$T_{FD}∼\left\{ \begin{array}{l} N\left(N\sigma^{2},N\sigma^{4}\right)\;\;H_{0}\\ N\left((1+\gamma )N\sigma^{2},(1+2\gamma )N\sigma^{4}\right)\;H_{1} \end{array}\right..$$

Let the threshold be $\eta_{FD}$. The false alarm probability and detection probability can be expressed as follows:(58)$$P_{f}=\mathrm{Pr}\left(T_{FD}\geq \eta_{FD}\right)=Q\left(\frac{\eta_{FD}-N\sigma^{2}}{\sqrt{N}\sigma^{2}}\right)\;,$$
(59)$$P_{d}=\mathrm{Pr}\left(T_{FD}>\eta_{FD}|H_{1}\right)=Q\left(\frac{Q^{-1}(P_{f})-\gamma \sqrt{N}}{\sqrt{1+2\gamma }}\right).$$
where $\eta_{FD}=\left(Q^{-1}(P_{f})\sqrt{N}+N\right)\sigma^{2}$.

#### 3.2.2. Matched Filter Detection

The main advantage of matched filtering is the short time to achieve a certain false alarm probability or detection probability. Hence, it requires perfect knowledge of the signal. In the time domain, the detection statistic of matched filtering is
(60)$$T_{MFD}=\sum_{n=0}^{L-1}s(n)x(n),$$
where s(n) is the transmitted signal, and x(n) is signal to be detector. Let $E=\sum_{n=0}^{L-1}x^{2}\left(n\right)$, i.e., all energy of the signal, and $\eta_{MFD}$ is the threshold. The false alarm probability and detection probability can be expressed as follows:(61)$$P_{f}=\mathrm{Pr}\left(T_{MFD}\geq \eta_{MFD}|H_{0}\right)=Q\left(\frac{\eta_{MFD}}{\sigma \sqrt{E}}\right)\;,$$
(62)$$P_{d}=\mathrm{Pr}\left(T_{MFD}\geq \eta_{MFD}|H_{1}\right)=Q\left(\frac{\eta_{MFD}-E}{\sigma \sqrt{E}}\right).$$
and $\eta_{MFD}=Q^{-1}(P_{f})\sigma \sqrt{E}$.

## 4. Experiments

For this section, we verified and compared the detection performances of the FPSED, FSED, and MTD discussed in Section 3 through Monte Carlo simulations. The primary signal was a binary phase shift keying(BPSK) modulated signal with symbol ratio 1 kbit/s, carrier frequency 1000 Hz, and sampling frequency $10^{5}$ Hz.

We performed all Monte Carlo simulations for at least $10^{4}$ independent trials. We set $P_{f}$ to 0.05. We used mean-square error (MSE) to measure the deviation between the theoretical values and actual statistical results. All the programs in the experiment were run in MATLAB set up on a laptop with a Core i5 CPU and 16GB RAM.

Since the PSER entropy $\mu_{1}$, and the sample entropy variance $\sigma_{0}^{2}$ and $\sigma_{1}^{2}$ cannot be calculated, a large number of simulation data were generated in the experiment to obtain ${\hat{\mu }}_{1}$, ${\hat{\sigma }}_{0}^{2}$, and ${\hat{\sigma }}_{1}^{2}$, which replace $\mu_{1}$, $\sigma_{0}^{2}$, and $\sigma_{1}^{2}$, respectively.

### 4.1. Experiments under $H_{0}$

This section verifies whether the calculation results of each statistic is correct by comparing the statistical results with the theoretical calculation result. The effects of noise intensity, the number of spectral lines and the number of intervals on interval probability, interval entropy, PSER entropy, and the variance of sample entropy are analyzed.

#### 4.1.1. Influence of Noise

According to the probability density function of PSER, PSER has nothing to do with noise intensity. Therefore, noise intensity has no effect on each statistic under $H_{0}$. In the following experiment, the variance of noise has 10 values ranging from 0.1 to 1 at 0.1 intervals, N=512, and m=500.

In Figure 5 and Figure 6, the theoretical and actual values of interval probability and interval entropy under different noise intensities are compared. The results show that the theoretical values are in good agreement with the statistical values, and the noise intensity has no effect on interval probability. In the first few intervals (i < 7), the interval probabilities are large, so the interval entropies are also large, contributing more to the total entropy.

In Figure 7, the theoretical values of PSER entropies are compared with the actual values. It can be seen that the theoretical values are basically consistent with the actual values, but the actual values are slightly smaller than the theoretical values. The actual values do not change with noise intensity, indicating that noise intensity has no effect on PSER entropy.

Since the theoretical variance of sample entropy cannot be calculated, only the variation of the actual variance of sample entropy with noise intensity is shown in Figure 8. It can be seen that the variance of sample entropy does not change with the noise intensity.

The above experiments show that the actual statistical results are consistent with the theoretical calculation results, indicating that the calculation methods of interval probability, interval entropy, PSER entropy, and sample entropy variance determined in this paper are correct.

#### 4.1.2. Influence of N

In Figure 9, the effect of N on the interval probability is shown. When m is fixed, the larger N is, the larger $p_{0}$ is, and the interval probabilities in other intervals will be smaller. The reason for that is that the larger N is, the smaller the energy ratio of each line to the entire power spectrum.

In Figure 10, the effect of N on the interval entropy is shown. The larger N is, the smaller $H_{i}$.

In Figure 11, the effect of N on the interval entropy is shown. When m is fixed, the larger N is, the smaller H(m,N) becomes. When N is the same, the larger m is, the larger the PSER entropy is.

In Figure 12, the effect of N on the variance of the sample entropy is shown. When m is fixed, the larger N is, the smaller Var(Z) becomes.

#### 4.1.3. Influence of m

In Figure 13, the effect of m on the PSER entropy is shown. When N is fixed, the larger m is, the larger H(m,N) becomes.

In Figure 14, the effect of N on the variance of the sample entropy is shown. When N is fixed, the larger m is, the variance of sample entropy increases first, then decreases, and then increases slowly.

#### 4.1.4. The Parameters for the Next Experiment

After theoretical calculations and experimental statistical analysis, the PSER entropy and sample entropy variance used in the following experiments were obtained, as shown in Table 1.

### 4.2. Experiments under $H_{1}$

When N is fixed, the change of PSER entropy of the BPSK signal with noise under $H_{1}$ is shown in Figure 15. It can be seen that the PSER entropy of BPSK signal decreases gradually with the increase of SNR. When the SNR is less than −15 dB, the PSER entropy of noise and that of the BPSK signal are almost the same; therefore, it is impossible to distinguish between noise and the BPSK signal.

Since the PSER entropy of the BPSK signal is always less than that of noise, the threshold $\eta_{1}$ should be used in BPSK signal detection.

As can be seen from Figure 16, with the increase of SNR, the sample entropy variance of the BPSK signal first increases and then gradually decreases.

When m is fixed, the change of PSER entropy and sample entropy variance of BPSK signal with noise under $H_{1}$ is shown in Figure 17 and Figure 18. It can be seen that the PSER entropy of BPSK signal decreases gradually with the increase of SNR.

As can be seen from Figure 18, with the increase of SNR, the sample entropy variance of the BPSK signal first increases and then gradually decreases.

### 4.3. Comparison of Detection Performance

When N is 512, m is 200, 500 and 1000, respectively. The detection results of the BSPK signal are shown in Figure 19, Figure 20 and Figure 21.

In Figure 19, the actual false alarm probabilities fluctuate slightly and do not change with the SNR, which is consistent with the characteristics of constant false alarm.

It can be seen from Figure 20 and Figure 21 that the detection probability of the PSER entropy detector is obviously better than that of the FSED method when m is 1000. However, when m is 200, the detection performance is lower than that of FSED. There is no doubt that the detection performance of matched filtering is the best.

The MSEs of the actual and theoretical probabilities of these experiments are given in Table 2.

The deviation between the actual and theoretical detection probabilities was very small, which indicated that the PSER entropy detector was accurate.

When m is 500, N is 256, 512, and 1024, respectively. The detection results for the BSPK signal are shown in Figure 22, Figure 23 and Figure 24.

It can be seen from Figure 23 and Figure 24 that when m is fixed, a larger N does not necessarily imply a better detection performance. The detection probability when N is 1024 is lower than that when N is 512. However, the detection performance of the full spectrum energy detection method will improve with the increase of N.

The MSEs of the actual and theoretical probabilities of these experiments are given in Table 3. The deviations between the actual and theoretical detection probabilities was very small when was 512 or 1024, however were higher when N was 256.

## 5. Discussions

### 5.1. Theoretical Calculation of Statistics

In Section 2, we analyzed the computational time complexity of each statistic. When m and N are large, the theoretical values of some statistics, such as Var(Z) under $H_{0}$, $H_{i}$, $H_{i,j}$, H(m,N), and Var(Z) under $H_{1}$, cannot be calculated. This restricts the further analysis on the detection performance of the PSER entropy detector.

There are two ways to solve this problem: reducing the computational complexity and finding an approximate solution. Which way is feasible requires further study.

### 5.2. Experience of Selecting Parameters

The detection probability $P_{d}$ of PSER entropy detection is related to the number of intervals m, the number of power spectrum lines N, and the SNR of the spectrum lines. Since the mathematical expressions of many statistics are too complex, the influence of the three factors on $P_{d}$ cannot be accurately analyzed at present. Based on a large number of experiments, we summarize the following experiences as recommendations for setting parameters.

(1)m cannot be too small. It can be seen from Figure 20, that, if m is too small, then the detection performance of the PSER entropy detector will be lower than that of energy detector. We suggest that m≥500.(2)N must be close to m. It can be seen from Figure 23, that N is not better when bigger. A large number of experiments show that when N is close to m, the detection probability is good.(3)When N is fixed, m can be adjusted appropriately through experiments.

### 5.3. Advantages of the PSER Entropy Detector

When using PSER entropy detection, the noise intensity does not need to be estimated in advance, and prior information of the signal to be detected is not needed. Therefore, the PSER entropy detector is a typical blind signal detection method.

### 5.4. Further Research

The detection performance of the PSER entropy detector will be further improved if some methods are used to improve the SNR of signals. In future research, a denoising method can be used to improve the SNR, the Welch or autoregressive method can be used to improve the quality of power spectrum estimation, and multi-channel cooperative detection can be used to increase the accuracy of detection.

## 6. Conclusions

In this paper, the statistical characteristics of PSER entropy are derived through strict mathematical analysis, and the theoretical formulas for calculating the PSER entropy and sample entropy variance from pure noise and mixed signals are obtained. In the process of derivation, we do not use the classical method of approximating PSER entropy using differential entropy, but use interval division and the multinomial distribution to calculate PSER entropy. The calculation results of this method are consistent with the simulation results, which shows that this method is correct. This method is not only suitable for a large number of intervals, but also suitable for a small number of intervals. A signal detector based on the PSER entropy was created according to these statistical characteristics. The performance of the PSER entropy detector is obviously better than that of the classical energy detector. This method does not need to estimate the noise intensity or require any prior information of the signal to be detected, and therefore it is a complete blind signal detector.

The PSER entropy detector can not only be used in spectrum sensing, but also in vibration signal detection, seismic monitoring, and pipeline safety monitoring.

## Figures and Tables

**Figure 1 entropy-23-00448-f001:**
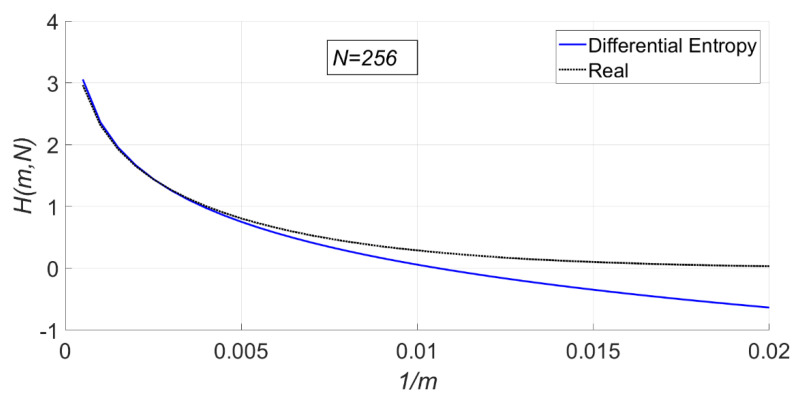
Comparison between the power spectrum subband energy ratio (PSER) entropy calcu-lated using differential entropy and the real PSER entropy.

**Figure 2 entropy-23-00448-f002:**
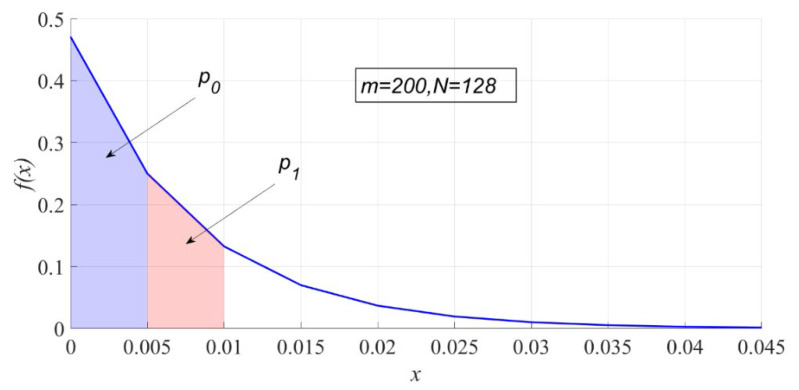
The interval probability under $H_{0}$.

**Figure 3 entropy-23-00448-f003:**
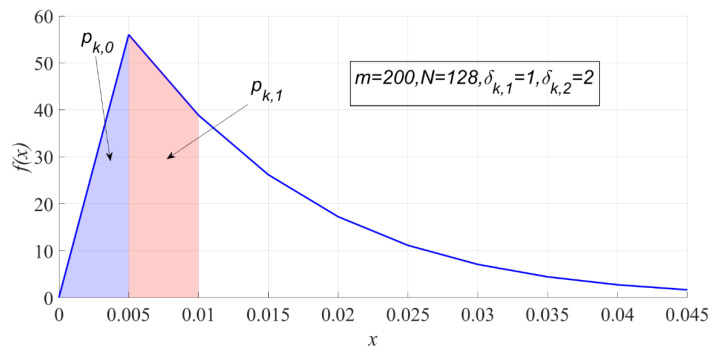
The interval probability of PSER under $H_{1}$.

**Figure 4 entropy-23-00448-f004:**
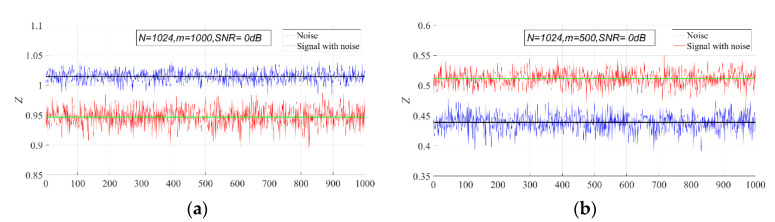
Comparison of the PSER entropy between GWN and the mixed signal (N = 1024, signal–noise ratio (SNR) = 0 dB). (**a**) m = 1000; (**b**) m = 500. The blue broken line is the sample entropy of GWN, and the red broken line is the sample entropy of the Ricker signal. The black line is the PSER entropy of GWN, and the blue line is the PSER entropy of the Ricker signal.

**Figure 5 entropy-23-00448-f005:**
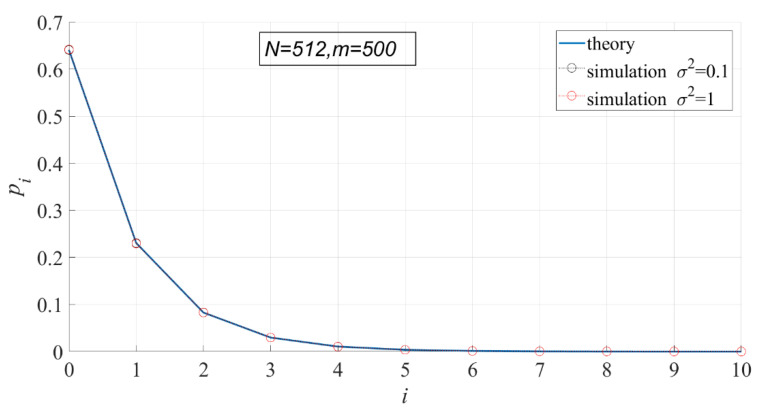
Comparison of the theoretical interval probability and the actual interval probability.

**Figure 6 entropy-23-00448-f006:**
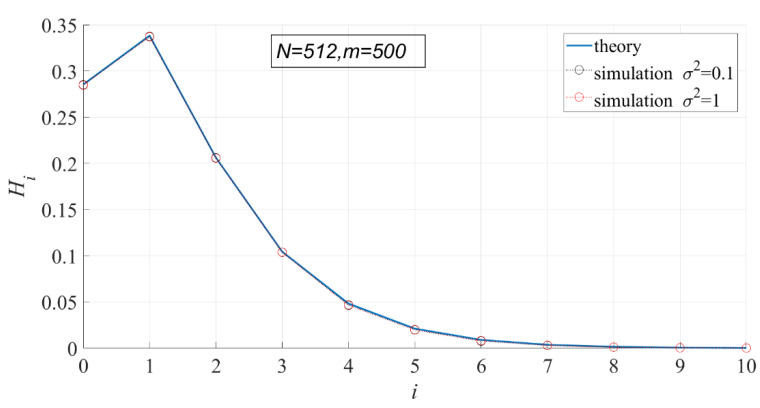
Comparison of the theoretical interval entropy and the actual interval entropy.

**Figure 7 entropy-23-00448-f007:**
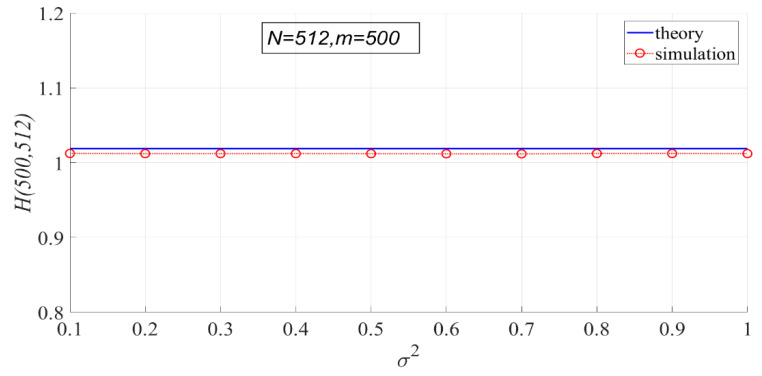
Comparison of theoretical PSER entropy and actual PSER entropy.

**Figure 8 entropy-23-00448-f008:**
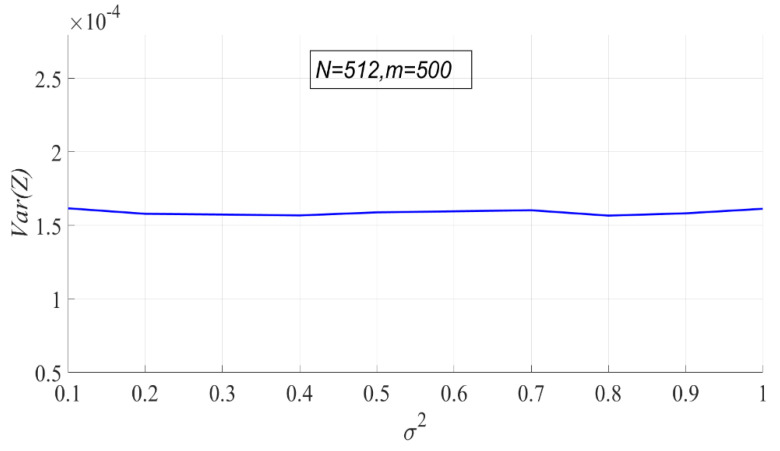
The actual variance of sample entropy.

**Figure 9 entropy-23-00448-f009:**
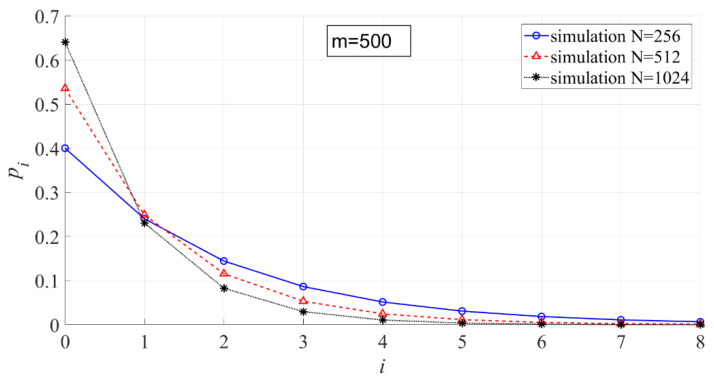
The effect of N on the interval probability (m = 500, N = 256, 512, 1024).

**Figure 10 entropy-23-00448-f010:**
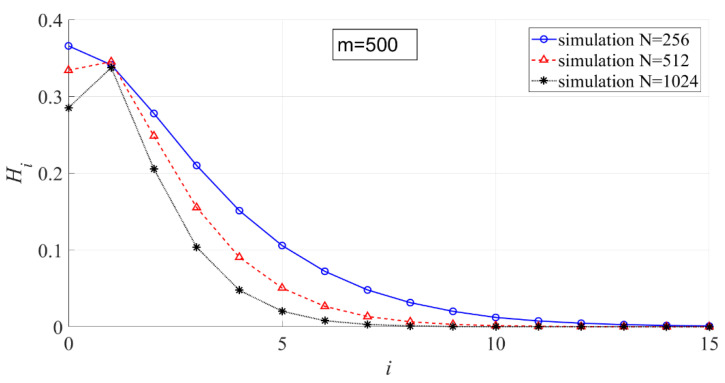
The effect of N on the interval entropy (m = 500, N = 256, 512, 1024).

**Figure 11 entropy-23-00448-f011:**
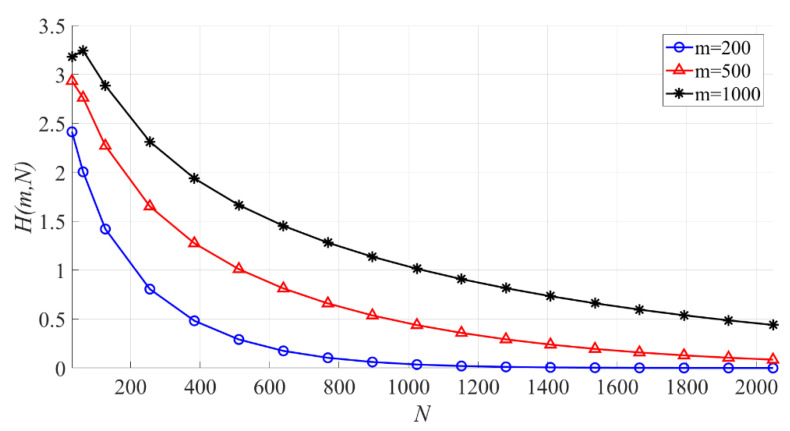
The effect of N on the PSER entropy (m = 200, 500, 1000).

**Figure 12 entropy-23-00448-f012:**
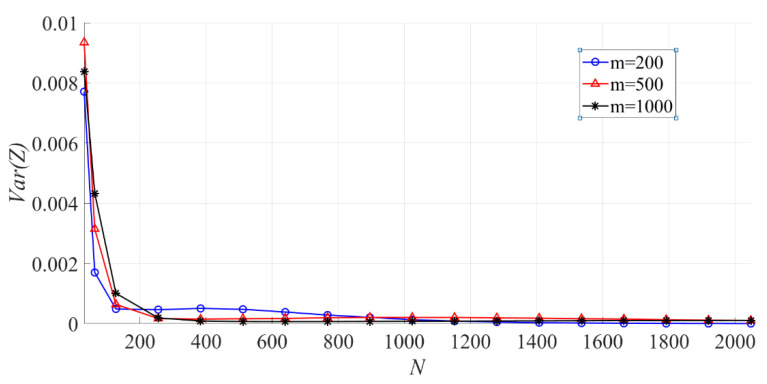
The effect of N on the variance of the sample entropy (m = 200, 500, 1000).

**Figure 13 entropy-23-00448-f013:**
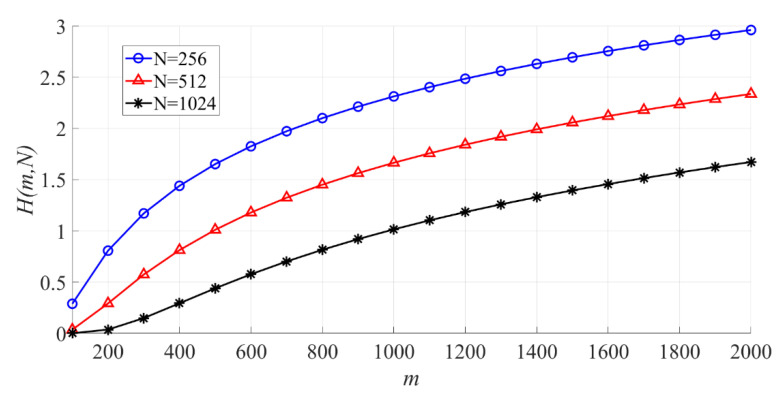
The effect of m on the PSER entropy (N = 256, 512, 1024).

**Figure 14 entropy-23-00448-f014:**
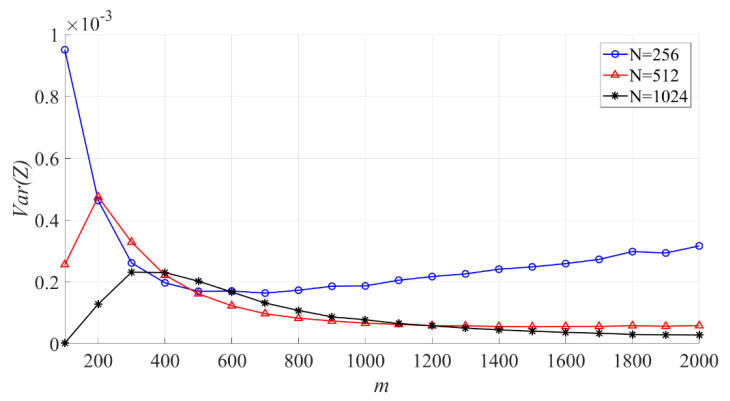
The effect of m on the variance of the sample entropy (N = 256, 512, 1024).

**Figure 15 entropy-23-00448-f015:**
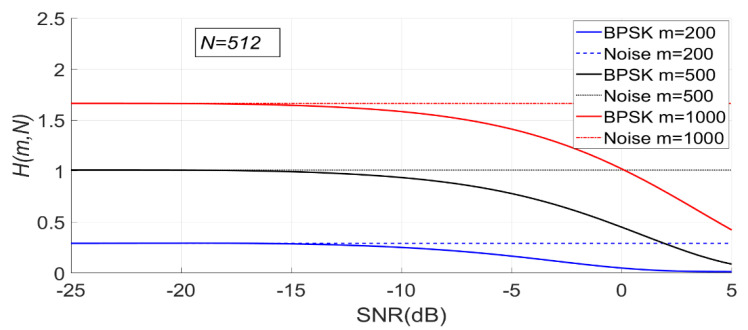
The change of PSER entropy of the binary phase shift keying (BPSK) signal with noise when N is fixed (m = 200, 500, 1000, N = 512).

**Figure 16 entropy-23-00448-f016:**
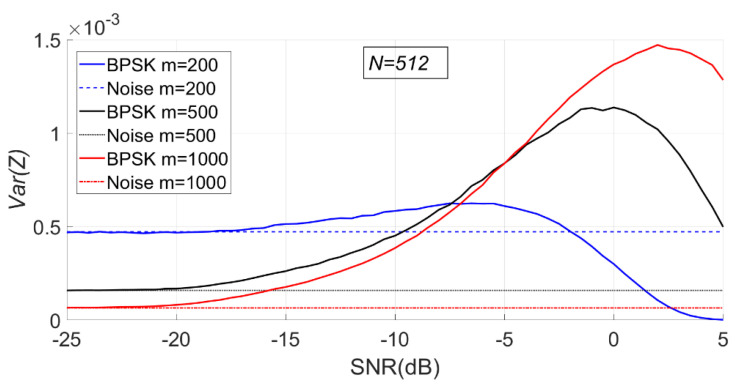
The change of sample entropy variance of BPSK signal with noise when N is fixed (m = 200, 500, 1000, N = 512).

**Figure 17 entropy-23-00448-f017:**
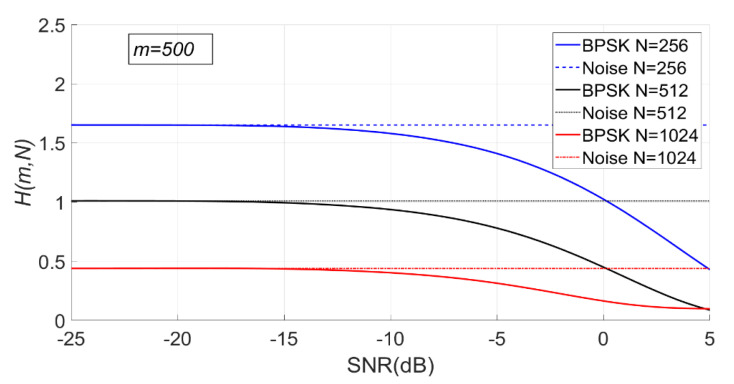
The change of PSER entropy of the BPSK signal with noise when m is fixed (m = 500, N = 256, 512, 1024).

**Figure 18 entropy-23-00448-f018:**
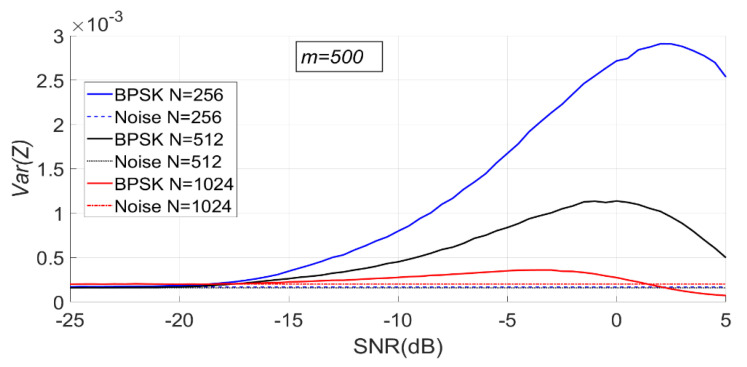
The change of sample entropy variance of the BPSK signal with noise when m is fixed (m = 500, N = 256, 512, 1024).

**Figure 19 entropy-23-00448-f019:**
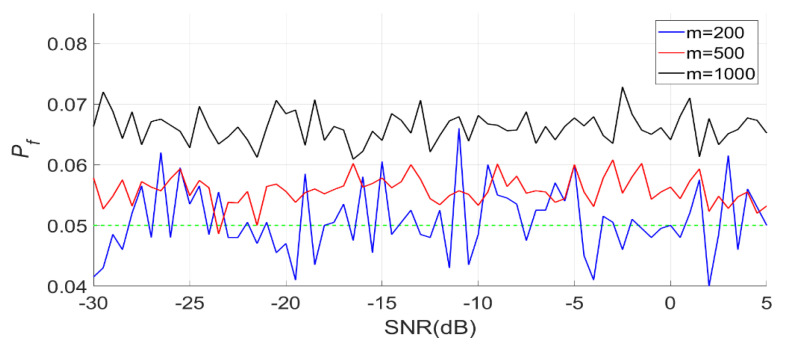
Actual false alarm probabilities of the PSER entropy detector (m = 200, 500, 1000, N = 512).

**Figure 20 entropy-23-00448-f020:**
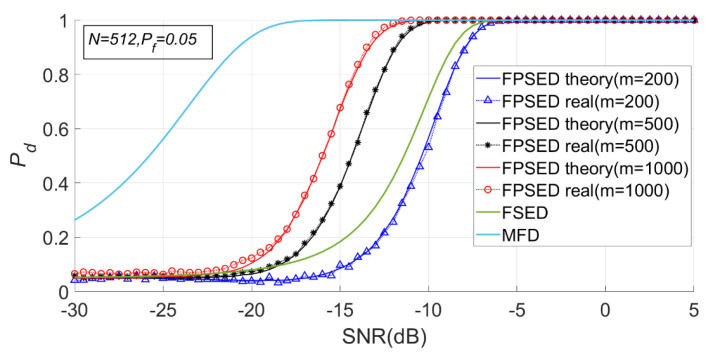
Detection probabilities of full spectrum energy detection (FSED), matched-filtering detector (MFD) and PSER entropy detectors (m = 200, 500, 1000, N = 512).

**Figure 21 entropy-23-00448-f021:**
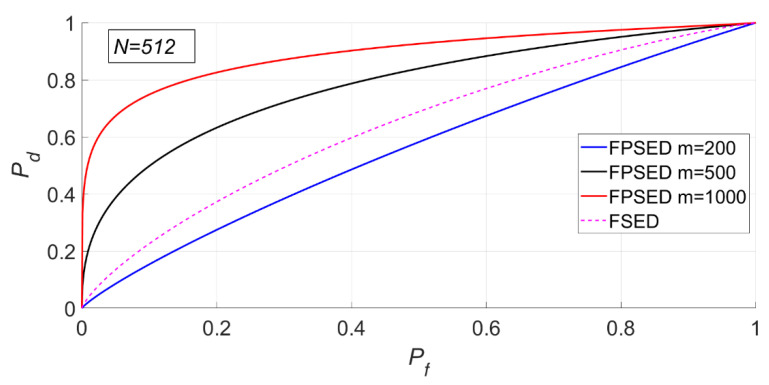
Receiver operating characteristic (ROC) curve of FSED and PSER entropy detectors (m = 200, 500, 1000, N = 512).

**Figure 22 entropy-23-00448-f022:**
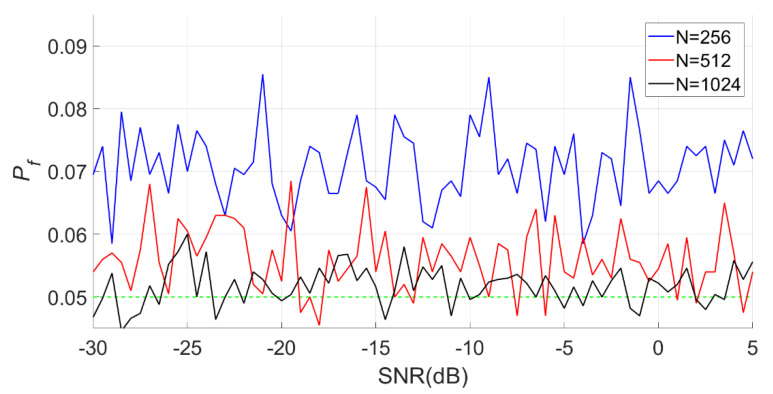
Actual false alarm probabilities of the PSER entropy detectors (m = 500, N = 256, 512, 1024).

**Figure 23 entropy-23-00448-f023:**
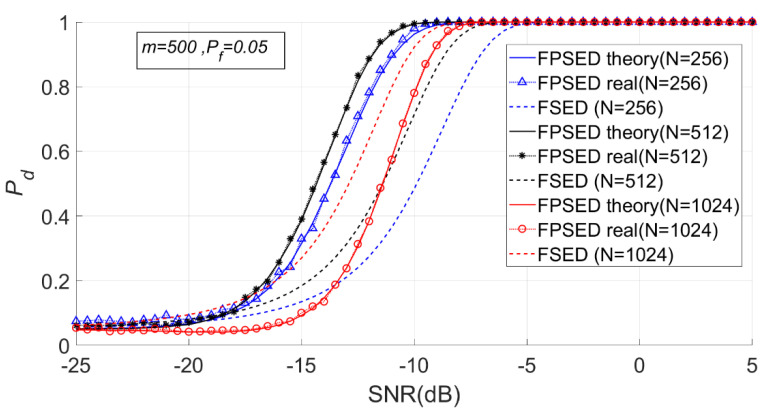
Detection probabilities of FSED and PSER entropy detectors (*m* = 500, N = 256, 512, 1024).

**Figure 24 entropy-23-00448-f024:**
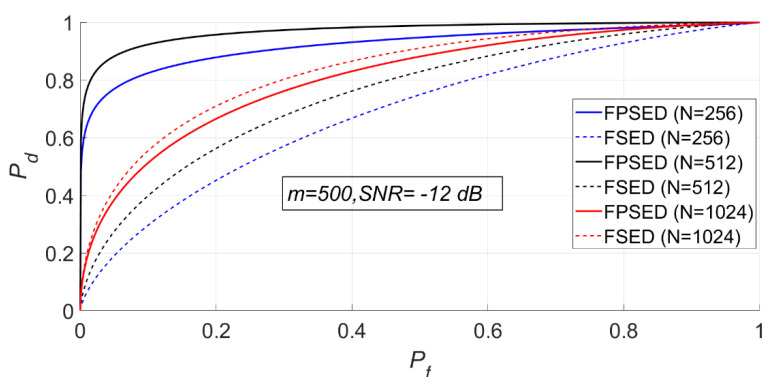
ROC of FSED and PSER entropy detectors (m = 500, N = 256, 512, 1024).

**Table 1 entropy-23-00448-t001:** The parameters under $H_{0}$.

	m	200	500	1000
N				
256		$\mu_{0}=0.809521$, ${\hat{\sigma }}_{0}^{2}=0.000453$	$\mu_{0}=1.652905$, ${\hat{\sigma }}_{0}^{2}=0.000169$	$\mu_{0}=2.313999$, ${\hat{\sigma }}_{0}^{2}=0.000192$
512		$\mu_{0}=0.291461$, ${\hat{\sigma }}_{0}^{2}=0.000466$	$\mu_{0}=1.011005$, ${\hat{\sigma }}_{0}^{2}=0.000159$	$\mu_{0}=1.665232$, ${\hat{\sigma }}_{0}^{2}=6.54\times 10^{-5}$
1024		$\mu_{0}=0.035906$, ${\hat{\sigma }}_{0}^{2}=0.000130$	$\mu_{0}=0.439132$, ${\hat{\sigma }}_{0}^{2}=0.000202$	$\mu_{0}=1.014594$, ${\hat{\sigma }}_{0}^{2}=7.7\times 10^{-5}$

**Table 2 entropy-23-00448-t002:** Mean-square errors (MSEs) between actual and theoretical probabilities (N = 512).

Probability	m = 200	m = 500	m = 1000
*P_f_*	0.3059 × 10^−4^	0.3887 × 10^−4^	2.6785 × 10^−4^
*P_d_*	0.5723 × 10^−4^	0.2777 × 10^−4^	1.1280 × 10^−4^

**Table 3 entropy-23-00448-t003:** MSEs between actual and theoretical probabilities (m = 500).

Probability	N = 256	N = 512	N = 1024
*P_f_*	4.687958 × 10^−4^	0.622887 × 10^−4^	0.125177 × 10^−4^
*P_d_*	2.067099 × 10^−4^	0.398899 × 10^−4^	0.145410 × 10^−4^

## Data Availability

All data is simulated.

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
