# Peer review of "A Novel Blind Signal Detector Based on the Entropy of the Power Spectrum Subband Energy Ratio"

_entropy, 2021, doi:10.3390/e23040448_

Round 1

Reviewer 1 Report

In this paper, we present a novel blind signal detector based on the entropy of the power spectrum subband energy ratio(PSER), the detection performance of which is significantly better than the classical energy detector. This detector is a full power spectrum detection method and does not require the noise variance or prior information about the signal to be detected. According to the analysis of the statistical characteristics of power spectrum subband energy ratio in the statistical intervals, this paper proposes concepts such as interval probability, interval entropy, sample entropy, joint interval entropy, PSER entropy and sample entropy variance.

  • The contribution is novel and has been very well explained in ABSTRACT and Introduction section. The literature review is nicely written; however, authors should summarize the literature discussion in a table to show the strengths and weaknesses of each method.
  • Authors have used several mathematical equations but has not cited the sources. The basic equations taken from other sources should be cited well. The author should avoid to include so much basic and well known math models. Instead they should focus on the contribution of this paper in derivations.
  • How mathematical models were simulated? For example results presented in Figure 4.
  • The authors should state the reason why there is a difference between two entropy values.
  • The details of the experimental setup, data collection, software/hardware interface should be presented.
  • There are a lot of impressive results depicted in various figures but authors need to describe the results in text as well.
  • A comparison table should be included to prove that the results are better than other similar studies.

Author Response

Point 1: The contribution is novel and has been very well explained in ABSTRACT and Introduction section. The literature review is nicely written; however, authors should summarize the literature discussion in a table to show the strengths and weaknesses of each method.

Response 1: The main weaknesses of entropy-based detection are that the variance of test statistics is unknown, which have been added to line 66-70. Therefore, the detection based on entropy has a major drawback in theory.

Point 2: Authors have used several mathematical equations but has not cited the sources. The basic equations taken from other sources should be cited well. The author should avoid to include so much basic and well known math models. Instead they should focus on the contribution of this paper in derivations.

Response 2: We have deleted some equations, and many equations are labeled with sources. But many equations are first presented.

Point 3: How mathematical models were simulated? For example, results presented in Figure 4.

Response 3: Figure 4 is the comparison between the PSER entropy calculated by Equation (14) and the real PSER entropy calculated by simulation experiment. The simulation experiment was described in detail in the revised paper. In fact, in this paper, the basic method of our research is the comparison of theory and simulation.

Point 4: The authors should state the reason why there is a difference between two entropy values.

Response 4: The reason was added in line 162-166.

Point 5: The details of the experimental setup, data collection, software/hardware interface should be presented.

Response 5: All experiments involve generating simulated data, calculating statistics values, and comparing them with theoretical values. These steps are relatively simple, so they were not described in the paper. There is no data collection. All the programs in the experiment are run in MATLAB set up a laptop with a Core i5 CPU and 16GB RAM.

Point 6: There are a lot of impressive results depicted in various figures but authors need to describe the results in text as well.

Response 6: We have added some descriptions for the figure in text, and some descriptions in 5.2.

Point 7: A comparison table should be included to prove that the results are better than other similar studies.

Response 7: The figures of detection probabilities and ROC between different detections can clearly compare the detection performance of each method. The two figures are the usual way to compare signal detection methods.

Reviewer 2 Report

This paper proposes a novel blind signal detector based on the entropy of the power spectrum subband energy ratio. The statistical characteristics of PSER entropy are derived through strict mathematical analysis, and the theoretical formulas for calculating the PSER entropy and sample entropy variance from pure noise and mixed signals are obtained. This method does not need to estimate the noise intensity or require any prior information of the signal to be detected, and therefore it is a complete blind signal detector, which proves its effectiveness and superiority. Overall this paper has a logical, intuitive flow and has detailed results. The paper fits well within the scope of the journal.

However, I have the following points that should be addressed before it is finally published in the journal.

  • The author presents the related methods about signal detector based on the entropy of the power spectrum subband energy ratio, but the literature analysis seems insufficient. In order to strengthen control motivation, it is better to enrich the literature review and summarize the main contributions in this point by point.

  • What are the main challenges for the blind signal detector based on the entropy? In this article, how did you solve this problem? Can you give some analyses for this problem?

  • The conclusion part can be expanded. In order to let readers better understand future work, please give specific research directions. It is suggested to read the following manuscripts: Towards Teaching by Demonstration for Robot-Assisted Minimally Invasive Surgery; An Incremental Learning Framework for Human-like Redundancy Optimization of Anthropomorphic Manipulators; A Smartphone-based Adaptive Recognition and Real-time Monitoring System for Human Activities.

Author Response

Point 1: The author presents the related methods about signal detector based on the entropy of the power spectrum subband energy ratio, but the literature analysis seems insufficient. In order to strengthen control motivation, it is better to enrich the literature review and summarize the main contributions in this point by point.

Response 1: There was no literature about the entropy of PSER, because it was first presented by this paper. We have introduced some articles on entropy-based detector with other metric, and we point out that these articles do not use the PSER entropy in the revised paper.

Point 2: What are the main challenges for the blind signal detector based on the entropy? In this article, how did you solve this problem? Can you give some analyses for this problem?

Response 2: The main challenges are that the variance of test statistics based on entropy is unknown, which have been added to line 66-70. In this article, we give a method to calculate the variance of the PSER, and this method can be used in the entropy of other metric.

Point 3: The conclusion part can be expanded. In order to let readers better understand future work, please give specific research directions. It is suggested to read the following manuscripts: Towards Teaching by Demonstration for Robot-Assisted Minimally Invasive Surgery; An Incremental Learning Framework for Human-like Redundancy Optimization of Anthropomorphic Manipulators; A Smartphone-based Adaptive Recognition and Real-time Monitoring System for Human Activities.

Response 3: We are not sure whether the PSER entropy detector is suitable for Human Activities. But it is more suitable for vibration signal detection, seismic monitoring and pipeline safety monitoring.

Round 2

Reviewer 1 Report

My comments have been addressed and the paper has been improved significantly.

Author Response

Thanks for your suggestions. But I think that  there is no suggestion for me this time?